# Modeling Method to Abstract Collective Behavior of Smart IoT Systems in CPS

**DOI:** 10.3390/s22135057

**Published:** 2022-07-05

**Authors:** Junsup Song, Dimitris Karagiannis, Moonkun Lee

**Affiliations:** 1Department of Computer Science and Engineering, Jeonbuk National University, Jeonju 54907, Jeonbuk, Korea; junsup@jbnu.ac.kr; 2Research Group Knowledge Engineering, University of Vienna, 1090 Vienna, Austria; dimitris.karagiannis@dke.univie.ac.at

**Keywords:** domain engineering, knowledge architecture, smart IoT, collective behavior, *n:2-Lattice*, dTP-Calculus, PRISM, SAVE, ADOxx Meta-Modeling Platform

## Abstract

This paper presents a new modeling method to abstract the collective behavior of Smart IoT Systems in CPS, based on process algebra and a lattice structure. In general, process algebra is known to be one of the best formal methods to model IoTs, since each IoT can be represented as a process; a lattice can also be considered one of the best mathematical structures to abstract the collective behavior of IoTs since it has the hierarchical structure to represent multi-dimensional aspects of the interactions of IoTs. The dual approach using two mathematical structures is very challenging since the process algebra have to provide an expressive power to describe the *smart* behavior of IoTs, and the lattice has to provide an operational capability to handle the state-explosion problem generated from the interactions of IoTs. For these purposes, this paper presents a process algebra, called *dTP-Calculus*, which represents the smart behavior of IoTs with non-deterministic choice operation based on probability, and a lattice, called *n:2-Lattice*, which has special *join* and *meet* operations to handle the state explosion problem. The main advantage of the method is that the lattice can represent all the possible behavior of the IoT systems, and the patterns of behavior can be elaborated by finding the traces of the behavior in the lattice. Another main advantage is that the new notion of equivalences can be defined within *n:2-Lattice*, which can be used to solve the classical problem of exponential and non-deterministic complexity in the equivalences of Norm Chomsky and Robin Milner by abstracting them into polynomial and static complexity in the lattice. In order to prove the concept of the method, two tools are developed based on the ADOxx Meta-Modeling Platform: SAVE for the dTP-Calculus and PRISM for the *n:2-Lattice*. The method and tools can be considered one of the most challenging research topics in the area of modeling to represent the collective behavior of Smart IoT Systems.

## 1. Introduction

The Cyber-Physical System (CPS) is one of the best implementation methods for IoT Systems, as shown in Figure 1, since the physical systems can be modeled, analyzed, and verified for safety at the time of design before construction activity [1].

However, since the systems consist of hundreds, thousands, or even tens of thousands of Smart IoTs, interacting with each other with communication and control while moving in some geographically distributed area, autonomously or heteronomously, with some critical missions, there are pressing needs to handle the size and complexity of the systems [2]. Particularly, the abstraction methods that handle the exponential growth problem of system states caused by interactions among IoTs in the systems, known as a state-explosion problem [3], must be presented [4].

In order to handle the problem, this paper presents a new modeling method to abstract the collective behavior of Smart IoT Systems, based on process algebra and a lattice.

Generally, it is known that process algebra can be one of the most suitable formal methods to model IoTs because each IoT can be represented as a process [5]. Similarly, it is known that a lattice can also be considered one of the most suitable mathematical structures to abstract the collective behaviors of IoTs because it has the hierarchical structure to represent multi-dimensional aspects of interactions of IoTs. (Note that the word *behavior* will be treated as a countable noun to distinguish an individual behavior from a group of behaviors.)

However, the dual approach using two mathematical structures in the paper is very challenging because the process algebra has to provide some expressive power to describe the *smart* behavior of IoTs, and the lattice has to provide some operational capability to handle the state-explosion problem generated by the interactions of IoTs.

In order to overcome the challenge, this paper presents a new process algebra, called *dTP-Calculus* [6,7], and a new lattice, called *n:2-Lattice* [8], as follows:*dTP-Calculus*: This process algebra can represent the *smart* behavior of IoTs in the systems with non-deterministic choice operation based on probability.*n:2-Lattice*: This lattice has two special *join* and *meet* operations to handle the state explosion problem caused by the interactions of IoTs in the systems.

The justification for the approach will be discussed with related works in Section 2.

The dual approach in the paper is implemented as follows with PRISM and SAVE tools, developed on the ADOxx Meta-Modeling Platform [9]:
(1)Phase I: Collective Behavior Modeling for Behavior Ontology with PRISM.The top box of Figure 2 shows the overview of the approach with PRISM, consisting of the following steps:
(i)Step 1: *Active Ontology* [10] is used to construct a class hierarchy of a domain, where all the actors in the domain, including their interactions, are depicted as classes and relations, respectively.(ii)Step 2: *Regular expression* is used to define all the collective behaviors of the domain, where each behavior is depicted as a sequence of interactions between actors. Further, their inclusion relations allow the organization of a hierarchical order, which forms a special lattice, namely, *n:2-Lattice*.(iii)Step 3: The behaviors are abstracted quantifiably with the notions of the cardinality and capacity of the actors.(iv)Step 4: Finally, a behavior ontology is defined by merging the *n:2-Lattices* into one single integrated lattice, based on common actors with the notion of the consistent quantification, for the domain.

The final lattice can be utilized for the interpretation of the collective behaviors of the systems for the following phases. This phase will be described in detail in Section 3.
(2)Phase II: Behavior Instance Extraction from SAVE [6,7,11].The bottom box of Figure 2 shows the specification and simulation method for a system with dTP-calculus, which are realized on the ADOxx Meta-Modeling Platform [9] as a tool, namely, SAVE, as follows:
(i)Step 1: Each IoT in the system is specified with dTP-Calculus for its operational requirements.(ii)Step 2: Each IoT in the system is simulated, and its output is generated.(iii)Step 3: Each abstract behavior instance is extracted from the output.

The simulation generates the abstract behavior instances of the system, which have to be interpreted for their collective behavioral patterns. This phase will be described in detail in Section 4.
(3)Phase III: Behavior Projection and Interpretation on Behavior Ontology with PRISM.The right box of Figure 2 shows that the behavior instances of the system can be projected to the behavior ontology for the system and interpreted for their collective behaviors in the patterns of the lattice after restructuring the regular behaviors into the abstract ones in the following steps:
(i)Step 1: Each abstract behavior instance is projected to Behavior Ontology.(ii)Step 2: The requirements for collective behavior instances are interpreted and verified.

This phase will be described in detail in Section 5.

In order to prove the concept of the approach, the authors developed the PRISM and SAVE tools on ADOxx and applied them to the Smart *Emergency Medical Service* (EMS) domain to abstract its collective behaviors. Further, a smart IoT example for the EMS Domain is also selected to interpret their collective behavior instances on PRISM by the described projection and interpretation steps.

The EMS example shows that the method can be evaluated to be effective in achieving the objectives and efficient in realizing the goals by constructing a hierarchy of collective behaviors in the lattice as the behavior ontology, as well as the following projection and interpretation tasks.

Further, the results show that, compared to other approaches [12,13,14,15,16], our method can be very innovative in representing the behaviors with collective patterns in the structure of the *n:2-Lattice*.

Further, the PRISM and SAVE tools demonstrate the efficiency and effectiveness of the approach for the feasibility of the method as the practical tools.

As a result, the main advantage of the method is that the lattice can represent all the possible behaviors of the IoT systems, and the patterns of behaviors can be elaborated by finding traces of the behaviors in the lattice.

Another main advantage is that the new notion of equivalence can be defined within the lattice, which can be used to solve the classical problem of exponential and non-deterministic complexity in the equivalences of Norm Chomsky and Robin Milner by abstracting them into polynomial and static complexity in the lattice. This will be discussed in detail in Section 5.2.

This paper is organized as follows. Section 2 discusses related works to justify the dual approach in the paper. Section 3 describes the modeling method for system behavior to construct the Behavior Ontology based on the *n:2-Lattice*. Section 4 describes the specification method for the systems with dTP-Calculus to generate the behavior instances through abstraction from their simulation. Section 5 shows the projection and interpretation method for the example on Behavior Ontology. Section 6 demonstrates the approach with the PRISM and SAVE tools as a proof of concept in the approach. Finally, Section 7 presents the conclusions and discusses future research.

## 2. Related-Works

### 2.1. Smart IoT and Process Algebra

Generally, an IoT System [17] is a system consisting of IoTs [18], which are computing objects with sensors interconnected through the Internet. Originally, the concept of an IoT was raised at the ITU (International Telecommunication Union) in 2005 [19] and became one of the future technologies provided by Cisco, Gartner, etc., between 2008 and 2009 [20,21]. The main characteristics of an IoT System are known to be its distributivity, mobility, communication (or interaction), real-time operation, etc. The most noticeable feature of the system is that the communication or interactions can be controlled by human intervention.

Compared with the IoT System, the main feature of a *Smart* IoT System is the capability of automation, which means that IoTs are able to communicate with each other and make their own decisions without human intervention, as the notion of “Smart” implies.

As stated, process algebra is one of the most suitable formal methods since each IoT can be modeled as a process. Further, the smartness can be represented as non-deterministic *choice* operations based on probability. Originally the non-deterministic choice operation was introduced by R. Milner for his PCCS [22] and followed by I. Lee for his PACSR [23]. However, their probabilities were based on simple conditional choice operations based on discrete probabilistic values without any distribution concepts, which implies that they are not suitable to express the smartness notion of Smart IoTs.

Recently, the following two process algebras with probabilistic choice operations were reported as follows:(1)pCCPS: It is a process algebra with non-deterministic choice operation based on discrete probability distribution only for CPS, whose definition is based on *probabilistic labelled transition semantics* (pLTS) [24]. Other probabilistic distributions are not defined yet.(2)PALOMA: Is another process algebra with choice operations based on exponential probability distribution to determine the location of a mobile agent [25]. Other probabilistic distributions are not yet defined.

Compared with the above process algebras, dTP-Calculus in the paper is able to represent most of the probability distribution models, which are supported by the underlining simulation facility of the ADOxx Meta-Modeling Platform for implementation of dTP-Calculus in the SAVE tool.

The comparative analysis of dTP-Calculus with other process algebra is shown in Table 1 from the perspective of the main features of the smart IoT system. Note that dTP-Calculus satisfies the basic requirements of the general IoT System with respect to distributivity, mobility, communication, and real-time, as well as the main requirement of the Smart IoT System with respect to probability.

### 2.2. State Explostion Problem and Abtraction

State explosion is the problem caused when the number of system states increases exponentially at the time of composition with other systems [26]. There were a number of approaches to handle the problem, but it is very hard to find any absolute solutions because the problem is caused by the nature of system composition. Most of the approaches focus on effective ways of reducing the number of system states in terms of abstraction.

Among these approaches, recent approaches related to process algebra can be summarized as follows:(1)Hierarchical approach: the flat level of the states of the process is hierarchically organized [27];(2)Grouping approach: A number of related states in the process are grouped together into a single abstract state at the time of composition [28];(3)Contextual Simplification: A set of specification contexts are abstracted into a single functional context [29].

Among these, the most recent approaches are:(1)A composition method that conjunctive and complement choice operations to reduce the size of the reachability graph, that is, the systems states in the graph [12];(2)Dividing method that a logical formula is divided into a number of sequential sub-formulas in order to apply model-checking [30].

Compared with the above approaches, the *n:2-Lattice* approach in the paper classifies the system states, in the beginning, into the categories with respect to the following notions:(1)Cardinality: It implies the composition patterns with respect to the number of actors for behaviors in the system;(2)Capacity: It implies the composition instances for the cardinality patterns.

Consequently, this approach reduces the states with respect to the types of behaviors, that is, a sequence of interactions among actors. Further, it represents the composition of two system states with respect to the same cardinality and capacity of the common actors. The comparative research was conducted and represented in [10].

## 3. Phase I: Collective Behavior Modeling for Behavior Ontology

This section presents a basic theory for behavior ontology as a knowledge architecture and the steps of the collective behavior modeling for the Smart EMS System Domain. The Smart EMS System is the system where, in case of emergency calls from patients, the patients from the emergency locations are transported to proper medical institutes by ambulances under the control of 911. The example will be demonstrated with PRISM in Section 6.

### 3.1. Theory: n:2-Lattice

We define a POSET ⟨L,≤⟩, where *a*, *b* are the members of Set *L*, while satisfying the following two conditions, to be *n*-*Lattice* ⟨L,≤,n⟩:(1)There exist more than one *join*s between *a*, *b* in Set {a, b}. That is, no *least upper bound* exists.(2)There exist more than one *meet*s between *a*, *b* in Set {a, b}. That is, no *greatest upper bound* exists.

By the above definition, n-Lattice is allowed to have multiple *join*s and *meet*s. This characteristic may violate the main property of the general lattice definition. However, it is possible to interpret it as a polymorphic property with respect to all of the possible binary relationships between two elements in the lattice.

Another important characteristic of the lattice is that the following two elements must exist in the lattice in order to control the multi-dimensional growth of *join*s and *meet*s:(1)*Super-Greatest Element* (SGE): As shown in Figure 3, SGE implies the biggest element among all of the elements of the *n*-Lattice.(2)*Super-Least Element* (SLE): Similarly, as shown in Figure 3, SLE implies the smallest element among all of the elements of the *n*-Lattice.

The implication of the multiple *join*s and *meet*s properties of the n-Lattice is that the exponential growth of the binary addition and binary multiplication is possible. Such exponential growth may cause the final structure of the *n*-Lattice to be uncontrollable.

SGE and SLE have the main characteristic that effectively controls the exponential growth of the *n*-Lattice caused by multiple *join*s and *meet*s. In addition, SGE and SLE satisfy the minimum requirements of the *n*-Lattice for the general lattice in the perspective of polymorphic structures.

Finally, we formally define the *n:2-Lattice*, based on the above characteristics: *n:2-Lattice* ⟨L,≤,n,2⟩ is defined as *n*-Lattice ⟨L,≤,n⟩ with both SGE and SLE.

A triclinic in *n:2-Lattice* is shown in Figure 4.

The *n:2-Lattice* can be considered as a lattice structure that has multiple *join*s and *meet*s for the internal elements of the lattice and has only one *join* and one *meet* for the bottom elements and the top elements of the lattice. It demonstrates the characteristics of the *n:2-Lattice* that allow non-determinism in the internal elements of the lattice, but it does not allow non-deterministic boundaries for the top and the bottom elements of the lattice.

### 3.2. Theory: Smart EMS Example

#### 3.2.1. Step 1: Active Ontology

The first step is to design an active ontology for the Smart EMS System Domain in the Smart IoT Systems. An active ontology consists of classes and subclasses in the domain, including their interactions.

The Active ontology of the Smart EMS System Domain is shown in Figure 5. The figure shows *Smart EMS System Domain* (EMS), *Ambulance* (A), *Patient* (P), *Place* (PL), *Location* (L), and *Hospital* (H) as classes or subclasses, and a1~a5 as interactions among classes and subclasses. The notions of the classes and their interactions are as follow:Classes:
(i)Smart EMS System Domain (EMS): Smart EMS System Domain, that is, EMS is the top-most class. It contains *Ambulance* (A), *Patient* (P), and *Place* (PL) as subclasses.(ii)Patient (P): Patient, that is, P, is a class that is transported to Hospital (H) by Ambulance (A).(iii)Ambulance (A): Ambulance, that is, A, is a class that transports Patient (P) to Hospital(H).(iv)Place (PL): Place, that is, PL, is a class that contains Location (A) and Hospital (H) as subclasses.(v)Location (L): Location, that is, L, a class that denotes the place where Patient (P) is at the beginning.(vi)Hospital (H): Hospital, that is, H, is a class that denotes the place where Patient (P) will be at the end.
Interactions:
(i)a_1_ = <A, L>: a1 is a movement action that Ambulance (A) drives to Location (L). It implies that the ambulance drives to the place where the patient is after receiving an emergency call from the patient.(ii)a_2_ = <P, A>: a2 is a movement, that is, take-on, action that Patient (P) performs onto Ambulance (A). It implies that the patient takes on the ambulance in order to go to the specific hospital.(iii)a_3_ = <A, H>: a3 is a movement that Ambulance (A) drives to Hospital (H). It implies that the ambulance drives to the hospital to transport the patient to the hospital.(iv)a_4_ = <A, P>: a4 is a movement, that is, take-off, action that Patient (P) performs off Ambulance(A). It implies that the patient takes off the ambulance at the hospital.(v)a_5_ = <P, H>: a5 is a movement action that Patient (P) moves into Hospital (H). It implies that the patient is now registered in the hospital for treatment.

#### 3.2.2. Step 2: Regular Behaviors

The collective behaviors are defined by determining their interactions, from Step 1, in sequence. Further, the quantified behaviors are classified into two types of behaviors: those with one single main actor and those with multiple main actors. Consequently, it is possible to have different views of different actors. Here an actor implies the lowest subclass from Step 1. For example, from the Smart EMS Domain, actors are *Ambulance* (A), *Patient* (P), *Location* (L), and *Hospital* (H). In addition, the behaviors of the actors are represented in the pattern of B(L, A, H, P). In the case of *Ambulance* (A) being the main actor, the behavior performed by one Ambulance can be represented by B(n, 1, n, n), and that of the multiple Ambulances can be achieved by B(n, n, n, n).

Here is a list of behaviors that can be defined for a single Ambulance in the regular expression:(1)$B_{1}=\langle a1,a2,a3,a4,a5\rangle$: An Ambulance drives to a Location (a1). a Patient at the Location takes on the Ambulance (a2). The Ambulance drives to a Hospital (a3). The Patient takes off the Ambulance (a4). The Patient is registered to the Hospital (a5).(2)$B_{2}=\langle a1,a2,a3,a4,a5\rangle_{}^{+}$: An Ambulance performs the behavior B_1_ repeatedly.(3)$B_{3}=\langle a1,\langle a2\rangle_{}^{+},a3,\langle a4,a5\rangle_{}^{+}\rangle_{}^{+}$: An Ambulance drives to a Location (a1). A number of Patients at the Location take on the Ambulance ($\langle a2\rangle_{}^{+}$). The Ambulance drives to a Hospital (a3). Each patient on the Ambulance takes off the Ambulance and is registered to the Hospital ($\langle a4,a5\rangle_{}^{+}$). The Ambulance performs this behavior repeatedly.(4)$B_{4}=\langle a1,\langle a2\rangle_{}^{+},\langle a3,a4,a5\rangle_{}^{+}\rangle_{}^{+}$: An Ambulance drives to a Location (a1). A number of Patients at the Location take on the Ambulance ($\langle a2\rangle_{}^{+}$). An Ambulance drives to each Hospital for each Patient, and the patient in the Ambulance takes off the Ambulance and is registered in the Hospital ($\langle a3,a4,a5\rangle_{}^{+}$). The Ambulance performs this behavior repeatedly.(5)$B_{5}=\langle a1,\langle a2\rangle_{}^{+},\langle a3,\langle a4,a5\rangle_{}^{+}|a3,a4,a5\rangle_{}^{+}\rangle_{}^{+}$: An Ambulance performs Behavior B_3_ or B_4_ repeatedly.(6)$B_{6}=\langle \langle a1,a2\rangle_{}^{+},a3,\langle a4,a5\rangle_{}^{+}\rangle_{}^{+}$: An Ambulance drives to a number of Locations for multiple Patients ($\langle a1,a2\rangle_{}^{+}$). The Ambulance drives to a Hospital (a3). Each patient in the Ambulance takes off from the Ambulance and is registered in the Hospital ($\rangle a4,a5\rangle_{}^{+}$). The Ambulance performs this behavior repeatedly.(7)$B_{7}=\langle \langle a1,a2\rangle_{}^{+},\langle a3,a4,a5\rangle_{}^{+}\rangle_{}^{+}$: An Ambulance drives to a number of Locations for multiple Patients ($\langle a1,a2\rangle_{}^{+}$). An Ambulance drives to each Hospital for each Patient, and the patient in the Ambulance takes off from the Ambulance and is registered in the Hospital ($\langle a3,a4,a5\rangle_{}^{+}$). The Ambulance performs this behavior repeatedly.(8)$B_{8}=\langle \langle a1,a2\rangle_{}^{+},\langle a3,\langle a4,a5\rangle_{}^{+}|a3,a4,a5\rangle_{}^{+}\rangle_{}^{+}$: An Ambulance performs Behavior B_6_ or B_7_ repeatedly.(9)$B_{9}={\left(\begin{array}{c} \left\langle a1,\langle a2\rangle_{}^{+},{\left\langle \begin{array}{c} a3,\langle a4,a5\rangle_{}^{+}|\\ a3,a4,a5 \end{array}\right\rangle }_{}^{+}\rangle \right|\\ \left\langle \langle a1,a2\rangle_{}^{+},{\left\langle \begin{array}{c} a3,\langle a4,a5\rangle_{}^{+}|\\ a3,a4,a5 \end{array}\right\rangle }_{}^{+}\right\rangle \end{array}\right)}_{}^{+}$: An Ambulance performs Behavior B_5_ or B_8_ repeatedly.

#### 3.2.3. Step 3: Abstract Behaviors

This step abstracts the regular behaviors from Step 2. Both the number of main actors and the numbers of their collaborating actors determine the degree of their interactions.

The notational format of *Abstract Behavior* is represented by $B\left(c_{\left\langle a_{1},\cdots ,a_{i}\right\rangle }^{i},\cdots ,c_{\left\langle z_{1},\cdots ,z_{k}\right\rangle }^{k}\right)$. Here in $c_{\left\langle a_{1},\cdots ,a_{i}\right\rangle }^{i}$ of the format, *c* represents an actor, where the upper and lower subscripts represent *cardinality* and *capacity*, respectively. Note that cardinality implies the number of the main actor, and that capacity implies the numbers of other actors that can get involved in the interaction. For example, $A_{\left\langle 1\right\rangle }^{1}$ implies that one Ambulance can hold one Patient, and $A_{\left\langle 1,1\right\rangle }^{2}$ implies that each of the two Ambulances can hold one Patient for each Ambulance.

The EMS example contains the following abstract behaviors for one Ambulance from Step 2:
(1)$B_{1}=B_{1}\left(P_{\left\langle 1\right\rangle }^{1},A_{\left\langle 1\right\rangle }^{1},H_{\left\langle 1\right\rangle }^{1}\right)$(2)$B_{2}=B_{2}\left(P_{\left\langle x_{1},\cdots ,x_{i}\right\rangle }^{i},A_{\left\langle 1\right\rangle }^{1},H_{\left\langle z_{1},\cdots ,z_{k}\right\rangle }^{k}\right)$(3)$B_{3}=B_{3}\left(P_{\left\langle x\right\rangle }^{1},A_{\left\langle y\right\rangle }^{1},H_{\left\langle z\right\rangle }^{1}\right)$(4)$B_{4}=B_{4}\left(P_{\left\langle x\right\rangle }^{1},A_{\left\langle y\right\rangle }^{1},H_{\left\langle 1_{1},\cdots ,1_{k}\right\rangle }^{k}\right)$(5)$B_{5}=B_{5}\left(P_{\left\langle x\right\rangle }^{1},A_{\left\langle y\right\rangle }^{1},H_{\left\langle z_{1},\cdots ,z_{k}\right\rangle }^{k}\right)$(6)$B_{6}=B_{6}\left(P_{\left\langle 1_{1},\cdots ,1_{i}\right\rangle }^{i},A_{\left\langle y\right\rangle }^{1},H_{\left\langle k\right\rangle }^{1}\right)$(7)$B_{7}=B_{6}\left(P_{\left\langle 1_{1},\cdots ,1_{i}\right\rangle }^{i},A_{\left\langle 1\right\rangle }^{1},H_{\left\langle 1_{1},\cdots ,1_{k}\right\rangle }^{k}\right)$(8)$B_{8}=B_{8}\left(P_{\left\langle 1_{1},\cdots ,1_{i}\right\rangle }^{i},A_{\left\langle y\right\rangle }^{1},H_{\left\langle z_{1},\cdots ,z_{k}\right\rangle }^{k}\right)$(9)$B_{9}=B_{9}\left(P_{\left\langle x_{1},\cdots ,x_{i}\right\rangle }^{i},A_{\left\langle y\right\rangle }^{1},H_{\left\langle z_{1},\cdots ,z_{k}\right\rangle }^{k}\right)$

Further, the example also contains the following abstract behaviors for *n* Ambulances:
(1)$B_{11}=B_{11}\left(P_{\left\langle x\right\rangle }^{1},A_{\left\langle y_{1},\cdots ,y_{j}\right\rangle }^{j},H_{\left\langle z\right\rangle }^{1}\right)$(2)$B_{12}=B_{12}\left(P_{\left\langle x\right\rangle }^{1},A_{\left\langle y_{1},\cdots ,y_{j}\right\rangle }^{j},H_{\left\langle z_{1},\cdots ,z_{k}\right\rangle }^{k}\right)$(3)$B_{13}=B_{13}\left(P_{\left\langle x_{1},\cdots ,x_{i}\right\rangle }^{i},A_{\left\langle y_{1},\cdots ,y_{j}\right\rangle }^{j},H_{\left\langle z\right\rangle }^{1}\right)$(4)$B_{14}=B_{14}\left(P_{\left\langle x_{1},\cdots ,x_{i}\right\rangle }^{i},A_{\left\langle y_{1},\cdots ,y_{j}\right\rangle }^{j},H_{\left\langle z_{1},\cdots ,z_{k}\right\rangle }^{k}\right)$

#### 3.2.4. Step 4: Behavior Lattice (BL) and Behavior Ontology (BO)

Lattice *L*_1_ can be constructed from Step 3 based on their inclusion relations among behaviors, as follows:

(1) $B_{1}⊑B_{2}$, (2) $B_{1}⊑B_{3}$, (3) $B_{1}⊑B_{4}$, (4) $B_{3}⊑B_{5}$

(5) $B_{4}⊑B_{5}$, (6) $B_{1}⊑B_{6}$, (7) $B_{1}⊑B_{7}$, (8) $B_{6}⊑B_{8}$

(9) $B_{7}⊑B_{8}$, (10) $B_{2}⊑B_{9}$, (11) $B_{5}⊑B_{9}$, (12) $B_{8}⊑B_{9}$

Similarly, Lattice *L*_n_ can be constructed from Step 3 based on their inclusion relations among behaviors, as follows:

(1) $B_{11}⊑B_{12}$, (2) $B_{11}⊑B_{13}$, (3) $B_{12}⊑B_{14}$, (4) $B_{13}⊑B_{14}$

Figure 6 shows one lattice of the behavior ontology for the example, merged from two lattices for one Ambulance at the bottom and for *n* Ambulances at the top.

This step includes merging the lattices into one integrated lattice of lattices, called Behavior Ontology, as shown in Figure 6.

## 4. Phase II: Behavior Instance Extraction

This section describes the method of extracting behavior instances for a target IoT system in the Smart EMS System Domain with dTP-Calculus in the SAVE tool.

### 4.1. Theory: dTP-Calculus

dTP-Calculus is a new process algebra designed to model distributed mobile real-time systems. For Smart IoT Systems, it can be used to model each IoT as a sequence of actions and interactions as a process. It was extended from dT-Calculus [11] by defining timed movements of processes with probability. The new feature includes the probabilistic choice operation, determined by the various probability distributions.

#### 4.1.1. Main Characteristics

dTP-Calculus provides the main characteristics of *mobility*, *synchronization*, *priority*, *time*, and *probability*, as follows.

##### Mobility

dTP-Calculus represents the movements of a process with respect to the type of movement mode and direction:(1)The movement mode: The mode is determined by the autonomy or heteronomy of the movement as follows:
(i)*Active* movements: The movements that a process performs autonomously.(ii)*Passive* movements: The movements that a process is performed heteronomously by other processes.
(2)The movement direction: The direction is determined by the target of the movement to or from a process:
(i)*Move-in* direction: The direction that a process moves into another process area.(ii)*Move-out* direction: The direction that a process moves out of another process area.

Table 2 shows the four types of movements available in dTP-Calculus.

##### Synchronization

The movement in dTP-Calculus is basically synchronous. Therefore, a handshake, known as permission, is necessary for both active and passive movements. Further, an asynchronous movement is also possible, determined by priorities in the form of an exception to the synchronous case. For example, a process with a higher priority can move in or out of other processes without any permission; it depends on its protocol.

##### Priority

Priority is a property that can be imposed on a process. It can be used for asynchronous communication and movement to handle exceptional situations in given protocols.

##### Time

The time for dTP-Calculus is discrete and represented by a natural number. The temporal properties of an action, i.e., communication or movements, can be defined as follows:(1)Ready Time: The minimum time needed for a process to prepare for an action.(2)Timeout: The maximum time needed for a process to prepare for an action.(3)Execution Time: The time needed for a process to perform an action itself.(4)Deadline: The time for a process to terminate an action including its ready time.(5)Period: The temporal period in which a process repeats an action.

##### Probability

dTP-Calculus defines the following probability distribution models for the probabilistic choice:(1)Discrete distribution;(2)Normal distribution;(3)Exponential distribution;(4)Uniform distribution.

The detailed definitions for the models are described in Section 4.1.2. The implementation of the models for dTP-Calculus is feasible due to the ADOxx meta-modeling platform since the platform provides the basic features and functionalities of the statistical simulation based on the different statistical models, such as R [31] and SAS [32].

#### 4.1.2. Syntax

Figure 7 shows the basic syntax of dTP-calculus. Each syntax is defined as follows:(1)*Action*: It denotes an operation performed by a process. There are four different types of action: null (*Empty*), communication (*Send*/*Receive*), movement (*Movement*), and control (*Control*).(2)*Timed action*: It is an action with the properties of *ready time*(*r*), *timeout*(*to*), *execution time*(*e*), and *deadline*(*d*). A detailed description of the properties is presented in Section Time. The types of Timed Action are the same as those of the actions in (1).(3)*Timed process*: It is a process with the same properties as Timed Action in (2).(4)*Priority*: It denotes the priority of a process. It is represented with a natural number. The higher value represents the higher priority, with the exception that 0 represents the highest priority.(5)*Nesting*: It denotes the inclusion relations among processes.(6)*Channel*: It denotes the communication channels between processes, which allow synchronization for communication.(7)*Choice*: It denotes the non-deterministic selection operation of actions or processes.(8)*Probabilistic choice*: It is the choice operation in (7) that is defined probabilistically, based on the following four probabilistic distribution models:
(i)*Discrete Distribution* (*D*): For discrete distribution, the probability is directly specified in the condition. For example, 0.7 and 0.3 are directly specified at *P* and *Q*, respectively, for the following probabilistic choice:
(1)$$P\left\{0.7\right\}{+}_{D}Q\left\{0.3\right\}$$(ii)*Normal Distribution* (N(μ,σ)): It is specified with the values of μ and σ. For example, 50 and 5 are specified for μ and σ, respectively, for the following probabilistic choice:(2)$$P\left\{v>52\right\}{+}_{N\left(50,5\right)}Q\left\{v\leq 52\right\}$$(iii)*Exponential Distribution* (Ex(λ)): It is specified with the value of λ. For example, 0.33 is specified for λ for the following probabilistic choice:(3)$$P\left\{v>2.5\right\}{+}_{E\left(0.33\right)}Q\left\{v\leq 2.5\right\}$$(iv)*Uniform Distribution* (U(l,u)): It is specified with the lower and upper bound values of l and u. For example, 3 and 7 are specified for l and u, respectively, for the following probabilistic choice:(4)$$P\left\{v>5\right\}{+}_{U\left(3,7\right)}Q\left\{v\leq 5\right\}$$(9)*Parallel*: It denotes that the multiple processes are running simultaneously.(10)*Exception*: It is the operation that is defined to handle an exception.(11)*Sequence*: It denotes an array of actions in a process, representing the basic patterns of actions in the process.(12)*Empty*: It denotes a *null* action, representing an idle process.(13)*Send/Receive*: It denotes a part of the paired communication actions, that is, sending or receiving, between two processes, based on synchronization.(14)*Movement request*: It denotes a request action for the synchronous movement among processes.(15)*Movement permission*: It denotes a permission action for the synchronous movement among processes.(16)*Create process*: It denotes a control action of a process to create its new child process inside itself.(17)*Kill process*: It denotes a control action of a process to terminate one of its internal processes with a lower priority.(18)*Exit process*: It denotes a control action for a process to terminate itself.

#### 4.1.3. Semantics

The semantics of dTP-Calculus are listed in Table 3 as a set of transition rules. The semantics of dTP-Calculus in the table is represented by the following form of the transition rules, where *Conclusion* can be derived from *Premise* when the *Side* condition is satisfied:(5)$$\frac{Premise}{Conclusion}\left(Side\;condition\right)$$

The premise and conclusion in this form can be represented as the following labelled transitions where the process state P can be transited to another process state $P^{\prime }$ with or without Action A.
(6)$$P\overset{\;\;}{\to }P^{\prime },\;P\overset{\;A\;}{\to }P^{\prime }$$

Each transition rule in the table can be defined as follows:
(1)*Sequence*: Defines that the proper execution of Action A makes the transition of A·P to P, without any premise and side condition.(2)*Choice: ChoiceL* and *ChoiceR* define that P or Q is selected for execution without any premise and side condition.(3)*Probability Choice*: It defines that *Choice* is performed probabilistically with a given premise and a side condition. For example, $A_{1}\left\{0.7\right\}+A_{2}\left\{0.1\right\}+A_{3}\left\{0.2\right\}$ implies that the probabilities for Actions $A_{1}$, $A_{2}$, and $A_{3}$ are 70%, 10%, and 20%, respectively.(4)*Com:* It defines the synchronous communication between P and Q on a channel with the conditions of $ch_{1}=ch_{2}$ and $msg_{1}=msg_{2}$. The *Send* action is defined by a message with overline ($\bar{msg1}$) and the *Receive* action is defined by a message without overline ($msg_{2}$). Synchronous communication is represented by the τ action.(5)*Parallel: ParallelL* and *ParallelR* define that the two processes, P and Q, are independent of each other and are executed in parallel. However, if they are dependent, the *ParallelCom Com* rule should be applied. It defines that if the two processes, P and Q, are synchronous actions, their τ action can occur synchronously in parallel, not affecting other processes.(6)*Nesting: NestingO* and *NestingI* define that the transition of P or Q does not affect the nested relation between P and Q. However, if there are synchronous actions between P and Q, the actions will affect both processes by their parallel synchronous transition as *NestingCom*. Note that the synchronous action between P and Q is denoted by the τ action.(7)*In*, *Out*, *Get*, and *Put: In* and *Get* are the moving-in actions of a process into another process, autonomously and heteronomously, respectively, and *Out* and *Put* are the moving-out actions of a process out of another process, autonomously and heteronomously, respectively. These actions are performed synchronously, meaning that the movement actions must be approved by the target process for both the moving-in and moving-out actions. Such synchronous movement actions are represented by δ action. Note that *In* and *Out* are active, and *Get* and *Put* are passive.(8)*InP*, *OutP*, *GetP*, and *PutP:* These rules are for asynchronous movements, represented by priorities. For example, if the process with a higher priority requests a movement to another process with a lower priority, there is no need to recieve permission from the other process. These rules can be used to handle some exceptional cases in emergency situations.(9)*TickTimeR:* It defines the elapsed local time in a process by decrementing the ready time r and the deadline d of an action by a time unit with ⊳1.(10)*TickTimeTO:* It defines the elapsed local time in a process by decrementing the timeout to and the deadline d of an action by a time unit with ⊳1 after the ready time r is completed in a condition where the synchronous partner process is not ready.(11)*TickTimeSyncE:* It defines the elapsed execution time for the synchronous action. If two synchronous actions A and $\bar{A}$ are ready, two actions are performed synchronously, and the execution time e and the deadline d of the actions are decremented by a time unit with ⊳1.(12)*TickTimeAsyncE:* It defines the elapsed execution time for the asynchronous action. Since the asynchronous does not require to wait for its timeout, to, it is possible to proceed to its execution just after its ready time, r. After that, its execution time, e, and deadline, d, are decremented by a time unit with ⊳1.(13)*TickTimeEnd:* It defines the termination of the action A by completing its execution time e. Note that ⊳1 implies the elapsed time unit.(14)*Timeout:* It defines the state of *Timeout error* at the time when *Timeout*(*to*) becomes 0 by the elapsed time unit with ⊳1, which implies a system fault. If an exception handler E is defined and the action with the fault is terminated, the handler E after the exception operator (\) is executed. Note that Process P is still valid.(15)*Deadline:* Similar to *Timeout*, defines the state of *Deadline error* at the time when *Deadline*(*d*) becomes 0 by the elapsed time unit with ⊳1, which implies a system fault. If an exception handler E is defined, the action with the fault is terminated and the handler process E after the exception operator (\) is executed. Process P is still valid.(16)*Period:* It defines the periodic repetition of Action A. In *Period*, Action A executes itself in the period A n time. It means that the value of n will be decremented by 1 after each ⊳per.(17)*Period End:* It defines the termination of the periodic repetition of Action A. Since the value of n is 0, Action A will not repeat itself any more after the elapsed unit period, ⊳per.

### 4.2. Smart IoT Example

A Smart IoT Example is selected to demonstrate the method to extract behavior instances of the example with dTP-Calculus and the SAVE tool. This subsection shows the steps of the extraction. The example will be demonstrated with PRISM in Section 6.

A Smart EMS Example consists of the following IoT instances for each of the EMS actors, as defined in Section 3.2:Patient: There is a total of eight Patients in the example: Each Patient is in Houses A, B, C, and D, respectively, and the other four Patients are in School E.Ambulance: There is a total of three Ambulances (A, B, and C) in the example.Place: There is a total of four Houses (A, B, C, and D) and one School(E) in the example.Hospital: There is a total of three Hospitals (A, B, and C) in the example.

Figure 8 shows a conceptual view of the system configuration.

#### 4.2.1. Step 1: Specification with dTP-Calculus

Figure 9 shows the code for the example in dTP-Calculus. There is a total of 21 processes as follows:• Control System: *CS*.• 911 Center: *911*.• Hospitals: *HospitalA*, *HospitalB*, *HospitalC*.• Ambulances: *AmbA*, *AmbB*, *AmbC*.• Places: *HouseA*, *HouseB*, *HouseC*, *HouseD*, *School*.• Patients: *PHBP1*, *PHBP2*, *PHBP3*, *PHD1*, *PHD2*, *PHD3*, *PFP1*, *PFP2*.

#### 4.2.2. Step 2: Simulation

The specifications for the example in dTP-Calculus are simulated by the Simulator of the SAVE tool. The snapshot of the output of the simulation for the example is shown in Figure 10. Note that the actions are highlighted, from which the behaviors are constructed based on the definitions from Section 3.2.

#### 4.2.3. Step 3: Extraction of Abstract Behavior Instances

This step extracts the abstract behavior instances of the example from the output of the simulation.

Regular behaviors are abstracted into abstract behaviors with respect to *cardinality* and *capacity*, which are defined as follows:Cardinality: The number of actors in an abstract behavior.Capacity: The possible number of other actors held by each actor in cardinality.

The abstraction information for the abstract behavior based on cardinality and capacity is classified into the following three levels:
Level 1: $A_{5\;\to \;\mathrm{The}\;\mathrm{Total}\;\mathrm{number}\;\mathrm{of}\;\mathrm{patients}\;}^{3\;\to \;\mathrm{The}\;\mathrm{total}\;\mathrm{number}\;\mathrm{of}\;\mathrm{Ambulances}}$Level 2: $A_{\langle 1,2,2\rangle \;\to \;\mathrm{The}\;\mathrm{number}\;\mathrm{of}\;\mathrm{patients}\;\mathrm{held}\;\mathrm{by}\;\mathrm{each}\;\mathrm{ambulances}}^{\langle 3\rangle \;\to \;\mathrm{The}\;\mathrm{total}\;\mathrm{number}\;\mathrm{of}\;\mathrm{Ambulances}}$Level 3: $A_{\langle \left[1\right],\left[2,3\right],\left[4,5\right]\rangle \;\to \;\mathrm{IDs}\;\mathrm{of}\;\mathrm{patients}\;\mathrm{held}\;\mathrm{by}\;\mathrm{each}\;\mathrm{ambulances}}^{\langle \left[1\right],\left[2\right],\left[3\right]\rangle \;\to \;\mathrm{ID}\;\mathrm{of}\;\mathrm{each}\;\mathrm{ambulances}}$

Next these six behavior instances can be abstracted into the following abstract behavior instances of the behavior ontology defined in Section 3.2:
(1)$B_{1.1}\left(P_{A^{H}}^{A},A_{\left\langle 1\right\rangle }^{A},H_{\left\langle 1\right\rangle }^{A}\right)=B_{1.1}\left(P_{\left\langle 1\right\rangle }^{1},A_{\left\langle 1\right\rangle }^{A},H_{\left\langle 1\right\rangle }^{1}\right)$(2)$B_{1.2}\left(P_{B^{H}}^{E},A_{\left\langle 1\right\rangle }^{A},H_{\left\langle 1\right\rangle }^{A}\right)=B_{1.2}\left(P_{\left\langle 1\right\rangle }^{1},A_{\left\langle 1\right\rangle }^{A},H_{\left\langle 1\right\rangle }^{1}\right)$(3)$B_{1.3}\left(P_{A^{B}}^{B},A_{\left\langle 1\right\rangle }^{B},H_{\left\langle 1\right\rangle }^{C}\right)=B_{1.3}\left(P_{\left\langle 1\right\rangle }^{1},A_{\left\langle 1\right\rangle }^{B},H_{\left\langle 1\right\rangle }^{1}\right)$(4)$B_{1.4}\left(P_{B^{B}}^{C},A_{\left\langle 1\right\rangle }^{B},H_{\left\langle 1\right\rangle }^{C}\right)=B_{1.4}\left(P_{\left\langle 1\right\rangle }^{1},A_{\left\langle 1\right\rangle }^{B},H_{\left\langle 1\right\rangle }^{1}\right)$(5)$B_{1.5}\left(P_{A^{D}}^{D},A_{\left\langle 1\right\rangle }^{B},H_{\left\langle 1\right\rangle }^{B}\right)=B_{1.5}\left(P_{\left\langle 1\right\rangle }^{1},A_{\left\langle 1\right\rangle }^{B},H_{\left\langle 1\right\rangle }^{1}\right)$(6)$B_{3.1}\left(P_{\left\langle A^{F},\;B^{F},C^{B}\right\rangle }^{E},A_{\left\langle 3\right\rangle }^{C},H_{\left\langle 3\right\rangle }^{C}\right)=B_{5.1}\left(P_{\left\langle 3\right\rangle }^{1},A_{\left\langle 3\right\rangle }^{C},H_{\left\langle 3\right\rangle }^{1}\right)$

Figure 11 pictorially shows the behavior $B_{1.1}\left(P_{A^{H}}^{A},A_{\left\langle 1\right\rangle }^{A},H_{\left\langle 1\right\rangle }^{A}\right)$. Note that behavior $P_{A^{H}}^{A}$ is represented with *P* as the Location, whose top and bottom subscripts imply Location *A*, that is, House *A*, and Patient *A* in House *A*, respectively. Note also that the top subscript of Patient *A* in Patient *A*($A^{H}$) implies the name of disease for the patient, that is, Heart Disesase.

The basic behavior for $B_{1.1}\left(P_{A^{H}}^{A},A_{\left\langle 1\right\rangle }^{A},H_{\left\langle 1\right\rangle }^{A}\right)$ is represented in the regular expression as $B_{1}=a1,a2,a3,a4,a5$, and behaves as follows:Ambulance *A* drives to House *A* (a1).Patient *A* takes on Ambulance *A* (a2).Ambulance *A* drives to Hospital *A* (a3).Patient *A* takes off Ambulance *A* (a4).Patient *A* is registered to Hospital *A* (a5).

The abstract behavior of $B_{1.1}\left(P_{A^{H}}^{A},A_{\left\langle 1\right\rangle }^{A},H_{\left\langle 1\right\rangle }^{A}\right)$ is $B_{1}\left(P_{\left\langle 1\right\rangle }^{1},A_{\left\langle 1\right\rangle }^{1},H_{\left\langle 1\right\rangle }^{1}\right)$, and it implies that one Ambulance transports one Patient to one Hospital.

These behavior instances are further abstracted with respect to the EMS behavior ontology defined in Section 3.2:(1)$B_{2.1}\left(P_{\left\langle 1,1\right\rangle }^{2},A_{\left\langle 1\right\rangle }^{A},H_{\left\langle 1,1\right\rangle }^{2}\right)⊑$$B_{9.1}\left(P_{\left\langle 1,1\right\rangle }^{2},A_{\left\langle 1\right\rangle }^{A},H_{\left\langle 1,1\right\rangle }^{2}\right)$(2)$B_{2.2}\left(P_{\left\langle 1,1,1\right\rangle }^{3},A_{\left\langle 1\right\rangle }^{B},H_{\left\langle 1,1,1\right\rangle }^{3}\right)⊑$$B_{9.2}\left(P_{\left\langle 1,1,1\right\rangle }^{3},A_{\left\langle 1\right\rangle }^{B},H_{\left\langle 1,1,1\right\rangle }^{3}\right)$(3)$B_{5.1}\left(P_{\left\langle 3\right\rangle }^{1},A_{\left\langle 3\right\rangle }^{C},H_{\left\langle 1,2\right\rangle }^{2}\right)⊑$$B_{9.3}\left(P_{\left\langle 3\right\rangle }^{1},A_{\left\langle 3\right\rangle }^{C},H_{\left\langle 1,2\right\rangle }^{2}\right)$

For example, $B_{2.1}\left(P_{\left\langle 1,1\right\rangle }^{2},A_{\left\langle 1\right\rangle }^{A},H_{\left\langle 1,1\right\rangle }^{2}\right)$ can be visualized in Figure 12. It represents an abstract behavior instance of $B_{2}$ for $B_{1.1}\left(P_{\left\langle 1\right\rangle }^{1},A_{\left\langle 1\right\rangle }^{A},H_{\left\langle 1\right\rangle }^{1}\right)$ and $B_{1.2}\left(P_{\left\langle 1\right\rangle }^{1},A_{\left\langle 1\right\rangle }^{A},H_{\left\langle 1\right\rangle }^{1}\right)$ for Abmulance *A*.

Further, all of the behavior instances of $B_{9.1}\left(P_{\left\langle 1,1\right\rangle }^{2},A_{\left\langle 1\right\rangle }^{A},H_{\left\langle 1,1\right\rangle }^{2}\right)$, $B_{9.2}\left(P_{\left\langle 1,1,1\right\rangle }^{3},A_{\left\langle 1\right\rangle }^{B},H_{\left\langle 1,1,1\right\rangle }^{3}\right)$, and $B_{9.3}\left(P_{\left\langle 3\right\rangle }^{1},A_{\left\langle 3\right\rangle }^{C},H_{\left\langle 3\right\rangle }^{1}\right)$ can be abstracted to $B_{14}$ for three Ambulances with a capacity of one, one, and three, as follows:(1)$B_{14}\left(P_{\left\langle 1,1,1,1,4\right\rangle }^{5},A_{\left\langle 1,1,3\right\rangle }^{3},H_{\left\langle 2,1,5\right\rangle }^{3}\right)$

This behavior instance can be visualized in Figure 13. It shows an abstraction behavior instance of all the mentioned $B_{9}$ behaviors.

## 5. Phase III: Behavior Projection and Interpretation on Behavior Ontology with PRISM

### 5.1. Projection of Behavior Instances to Behavior Ontology

Figure 14 shows the results of the projection of the behavior instances from the previous phase on the Behavior Ontology of the Smart EMS Systems Domain.

### 5.2. Interpretations for Equivalences

The strong and weak equivalences relations for the Smart EMS System Domain extracted from Figure 14 are as follows:(1)Strong equivalences:
(i)$B_{1.1}\left(P_{\left\langle 1\right\rangle }^{1},A_{\left\langle 1\right\rangle }^{A},H_{\left\langle 1\right\rangle }^{1}\right)∼B_{1.2}\left(P_{\left\langle 1\right\rangle }^{1},A_{\left\langle 1\right\rangle }^{A},H_{\left\langle 1\right\rangle }^{1}\right)$(1-1-a in Figure 14);(ii)$B_{1.3}\left(P_{\left\langle 1\right\rangle }^{1},A_{\left\langle 1\right\rangle }^{B},H_{\left\langle 1\right\rangle }^{1}\right)∼B_{1.4}\left(P_{\left\langle 1\right\rangle }^{1},A_{\left\langle 1\right\rangle }^{B},H_{\left\langle 1\right\rangle }^{1}\right)∼B_{1.5}\left(P_{\left\langle 1\right\rangle }^{1},A_{\left\langle 1\right\rangle }^{B},H_{\left\langle 1\right\rangle }^{1}\right)$(1-1-b in Figure 14);(iii)$B_{1.1}\left(P_{\left\langle 1\right\rangle }^{1},A_{\left\langle 1\right\rangle }^{A},H_{\left\langle 1\right\rangle }^{1}\right)∼B_{1.2}\left(P_{\left\langle 1\right\rangle }^{1},A_{\left\langle 1\right\rangle }^{A},H_{\left\langle 1\right\rangle }^{1}\right)∼B_{1.3}\left(P_{\left\langle 1\right\rangle }^{1},A_{\left\langle 1\right\rangle }^{B},H_{\left\langle 1\right\rangle }^{1}\right)∼\;B_{1.4}\left(P_{\left\langle 1\right\rangle }^{1},A_{\left\langle 1\right\rangle }^{B},H_{\left\langle 1\right\rangle }^{1}\right)∼B_{1.5}\left(P_{\left\langle 1\right\rangle }^{1},A_{\left\langle 1\right\rangle }^{B},H_{\left\langle 1\right\rangle }^{1}\right)$(1-2 in Figure 14);(iv)$B_{2.1}\left(P_{\left\langle 1,1\right\rangle }^{2},A_{\left\langle 1\right\rangle }^{A},H_{\left\langle 1,1\right\rangle }^{2}\right)∼B_{2.2}\left(P_{\left\langle 1,1,1\right\rangle }^{3},A_{\left\langle 1\right\rangle }^{B},H_{\left\langle 1,1,1\right\rangle }^{3}\right)$(2-1 in Figure 14);(v)$B_{9.1}\left(P_{\left\langle 1,1\right\rangle }^{2},A_{\left\langle 1\right\rangle }^{A},H_{\left\langle 1,1\right\rangle }^{2}\right)∼B_{9.2}\left(P_{\left\langle 1,1\right\rangle }^{2},A_{\left\langle 1\right\rangle }^{B},H_{\left\langle 1,1,1\right\rangle }^{3}\right)∼B_{9.3}\left(P_{\left\langle 3\right\rangle }^{1},A_{\left\langle 3\right\rangle }^{C},H_{\left\langle 3\right\rangle }^{1}\right)$(9-1 in Figure 14).(2)Weak equivalences:(i)$B_{1.1}\left(P_{\left\langle 1\right\rangle }^{1},A_{\left\langle 1\right\rangle }^{A},H_{\left\langle 1\right\rangle }^{1}\right)\approx B_{3.1}\left(P_{\left\langle 3\right\rangle }^{1},A_{\left\langle 3\right\rangle }^{C},H_{\left\langle 3\right\rangle }^{1}\right)$;(ii)(The above 1. iii) $\approx B_{3.1}\left(P_{\left\langle 3\right\rangle }^{1},A_{\left\langle 3\right\rangle }^{C},H_{\left\langle 3\right\rangle }^{1}\right)$;(iii)$B_{2.1}\left(P_{\left\langle 1,1\right\rangle }^{2},A_{\left\langle 1\right\rangle }^{A},H_{\left\langle 1,1\right\rangle }^{2}\right)\approx B_{5.1}\left(P_{\left\langle 3\right\rangle }^{1},A_{\left\langle 3\right\rangle }^{C},H_{\left\langle 1,2\right\rangle }^{2}\right)$;(iv)(The above 1. iv) $\approx B_{5.1}\left(P_{\left\langle 3\right\rangle }^{1},A_{\left\langle 3\right\rangle }^{C},H_{\left\langle 1,2\right\rangle }^{2}\right)$.

The types of the above equivalences are determined by the cardinality and capacity of the actor with respect to the Behavior Ontology based on the *n:2-Lattice*. Since all the possible compositional complexity of the interactions among the actors in the types of the above equivalences are already abstracted with respect to the collective patterns of their behaviors, it is not necessary to consider the non-deterministic states of the complexity existing in the interactions among the actors of the behaviors. Therefore, it is not necessary to consider the non-deterministic problem caused by Norm Chomsky’s equivalences and Robin Milner’s bisimulations. It could be one of the main advantages of the approach in the paper provided by the Behavior Ontology to analyze and interpret these types of equivalences.

### 5.3. Future Research for Probable Similarity

Further, probable similarity can be defined with respect to the strong and weak equivalences since dTP-Calculus allows the choice operations with probability. It implies that two systems can probably be similar with respect to a set of identical processes, or IoTs, with the same choice operations with different probabilities, under a tolerable condition of the probabilistically acceptable threshold in similarity.

## 6. Proof of Concepts

### 6.1. The PRISM Tool

The PRISM tool was realized on the ADOxx Meta-Modeling Platform. ADOxx is one of the best-known open SW and originated from the OMiLAB of the University of Vienna. It is recognized as one of the most innovative meta-modeling tools open to the public. In total more than 70 open models are available on ADOxx for non-profit public applications [33].

Figure 15 shows the architecture with modeling views of PRISM. The graphical representations of the models in PRISM are defined by the ADOxx Development Tool, and the procedures of its components are constructed using the ADOxx libraries. The detailed algorithms of the procedures are programmed with the ADOScript language.

The PRISM tool consists of the ADOxx Platform, PRISM Components, and PRISM Models, as follows:(1)ADOxx Platform: ADOxx is the platform supporting the meta-modeling method to implement the modeling language, mechanisms, and algorithms of PRISM. In terms of PRISM, ADOxx can be classified into the following sub-layers:
(i)First Sub-Layer: It consists of pre-defined functions to develop the PRISM modeling tool. The functions are used to implement the basic modeling language, mechanisms, and algorithms of the modeling tool.(ii)Second Sub-Layer: It consists of APIs provided by ADOxx. APIs are provoked by the ADOScripts language and used to implement the extended functions, that is, user-defined functions of the modeling tool.(iii)Third Sub-Layer: It consists of the ADOxx repository to store the products produced by the modeling tool, as well as the functions to export and import the products to/from external supporting systems.(2)PRISM Components: PRISM provides all the functions to model and analyze the behaviors stated in the steps in Section 3 and is supported by the following components:
(i)Regular Behavior Generator (RBG): RBG is used to construct the regular behaviors based on Active Ontology. The RB model is generated as a result.(ii)Abstract Behavior Generator (ABG): ABG is used to construct the abstract behaviors based on the RB model produced from the above (i). The AB model is generated at the end.(iii)Behavior Lattice Generator (BLG): BLG is used to construct the behavior lattice based on the AB model produced from the above (ii). The BL model is generated at the end.(iv)Behavior Lattice Merger (BLM): BLM is used to construct one behavior lattice integrated from the BL models of the two different systems. Note that integration is only possible when the two different BL models have a common main actor.(v)Behavior Interpreter (BI): BI is used to project the collective behavior patterns of the selected domain onto the BL mode produced above (iii). The iBL model is generated at the end. In order to utilize BI, it is necessary to collect the behavior data of the selected system domain example from the SAVE tool.(3)PRISM Modelers: PRISM Modelers consist of the products produced by the PRISM tool:
(i)Class Diagram (CD): The CD model is the model designed for the hierarchical structure of the classes in the system domain.(ii)Active Diagram (AD): The AD model is the model designed for the actions among the classes in the CD model from (1). Active ontology is constructed at the end.(iii)Regular Behavior (RB): The RB model consists of the regular behaviors generated from the AD model.(iv)Abstract Behavior (AB): The AB model consists of abstract behaviors generated from the RB model.(v)Behavior Lattice (BL): The BL model is generated from the AB model. The inclusion relations among abstract behaviors can be found in the model.(vi)Merged Lattice of Behavior Lattices (mLBL): The mLBL model is constructed from two different BL models.(vii)Interpreted Behavior Lattice (iBL): The iBL model is generated from the BL model. This model allows the extraction of the collective behavior patterns of the example from the selected system domain.

### 6.2. The SAVE Tool

Figure 16 shows the architecture with modeling views of SAVE (Specification, Analysis, Verification, and Evaluation). SAVE is the tool suite to specify, analyze, and verify the operational and safe requirements of systems with dTP-Calculus in Section 4. SAVE was developed on the ADOxx Meta-Modeling Platform. In addition, SAVE provides the graphic notations and simulation functions for specification, analysis, and verification, based on the meta-modeling methods of ADOxx. The basic components of SAVE are as follows:(1)Specifier: Is the modeling tool to specify the operational requirements of a system by using the graphic notations of dTP-Calculus. It consists of *In-the-Large* (ITL) and *In-the-Small* (ITS) modelers. ITL is the model to specify the system view consisting of processes and their inclusion relations, including communication channels, in a conceptual geographical space. It represents a process as a node, as well as inclusion relations. In addition, the channels, represented as arcs, are connecting processes for communication. ITS is the model to specify the process view consisting of the actions, communications, and movements of a process. ITS represents a process as a Process Lane, which consists of blocks of the actions, communications, and movements of the process. The specifier generates ITL and ITS models. The details are presented in [6,7].(2)Analyzer: It is the tool that simulates the execution paths generated using the ITL and ITS models from (1). As a result, an execution model is generated automatically. It analyzes all of the execution paths of the simulation model, and it represents the simulation results of each execution path pictorially. The details are presented in [6,7].(3)Verifier: It is the tool to verify the safety and security requirements of a system from the simulation results from (2). As a result, the geo-temporal space (GTS) model is generated. The GTS model represents the simulation results of all the actions and movements of the processes pictorially in a two-dimensional space. The details are presented in [6,7].

### 6.3. Smart EMS Example

#### 6.3.1. Phase I with PRISM

Step 1 of Phase I is to construct the target Active Ontology, consisting of classes and interactions. A class can include subclasses with inclusion relations. An interaction implies a movement of class between two classes. Figure 17 shows the active ontology of the Smart EMS modeled in PRISM. In the figure, EMS implies the upper class, which includes Ambulance (A), Patient (P), and Place (PL) as subclasses. Note that the Place (PL) class also includes Location (L) and Hospital (H) as subclasses. The figure also includes a1~a5 as interactions between classes.

Step 2 of Phase I is to define regular behaviors with a sequence of the interactions defined in Step 1. The regular behavior implies the behavior with the main class of the cardinality 1. Figure 18 shows the regular behaviors modeled in PRISM.

Step 3 of Phase I is to define the Abstract behaviors by abstracting the regular behaviors defined in Step 2. The Abstract behavior implies the behavior with the main class of the cardinality *n*. The Abstract behaviors and their inclusion relations are automatically generated from regular behaviors by PRISM in this step. Step 3 of Phase I is to construct a behavior ontology from the abstract behaviors and inclusion relations from Step 3. Figure 19 shows the behavior ontology in the lattice structures and the inclusion relations in a dialog box.

#### 6.3.2. Phase II with SAVE

Figure 20 shows the Smart EMS Example in SAVE. The circles on the left of the figure imply the locations where patients are. Similarly, regarding the figure, the ones in the middle and the ones on the right imply the ambulances and the hospitals.

Figure 21 shows the results of the simulation for the specification defined in Figure 20. During the simulation, the movements of all the processes, that is, Ambulances and Patients, along with the temporal perspective, that is, the system behaviors, are analyzed and synthesized, as shown in Figure 10. In addition, the raw data that will be used in Phase III are collected, as shown in Figure 22.

#### 6.3.3. Phase III with PRISM

Figure 23 shows the types of abstract behaviors of the ambulances as a main class from the raw data in Figure 22. For example, the types of abstract behaviors performed by Ambulance 1 are B1, B1, B2, and B9. Figure 24 and Figure 25 show the results of projecting the abstract behaviors performed by all the ambulances onto the behavior ontology generated during Phase I.

## 7. Conclusions and Future Research

This paper presented a new modeling method to abstract the collective behavior of Smart IoT Systems in CPS based on *dTP-Calculus* and *behavior ontology* and demonstrated its feasibility with two tools: PRISM and SAVE. The modeling method consisted of these phases:(1)Phase 1: Each behavior of the IoT Systems Domain was defined as a sequence of the interactions and/or movements of a group of the IoTs in the systems with respect to each type of IoT. Since interactions and movements among the behaviors were overlapped, the behaviors were organized in a lattice structure called *n*:2-*Lattice*, which has the special properties of multiple *join*s and *meet*s. Further, the lattice could be interpreted with respect to the cardinalities of the types of IoTs, and it was possible to construct a type-oriented knowledge architecture for all the possible collective behavior of the IoT systems.(2)Phase 2: An IoT Example from the IoT Systems Domain was modeled with dTP-Calculus, where all the actions of each IoT in the example were defined as the interactions and movements of processes with dTP-Calculus.(3)Phase 3: The output of the simulation was abstracted and projected to the Behavior Ontology of the domain.

The method demonstrated that dTP-Calculus was appropriate to model Smart IoTs in CPS and that the Behavior Ontology based on the *n:2-Lattice* had the structural capability to represent multi-dimensional aspects of behaviors in a hierarchical structure. Consequently, the combination of two mathematical structures allowed for the efficient and effective abstraction of the collective behavior of Smart IoT Systems in CPS.

The main advantage of the method is that the architecture can represent all the possible behaviors of IoT systems and that the patterns of behavior can be elaborated by finding traces of the behaviors in the lattice.

Another main advantage is that the new notion of equivalence can be defined within the behavior ontology, which can be used to solve the classical problem of exponential and non-deterministic complexity in the equivalences of Norm Chomsky and Robin Milner by abstracting them into polynomial and static complexity in the lattice. It means that the ontology provides a systematic mechanism to specify, analyze, and verify the equivalences based on a formal structure, that is, the *n*:2-*Lattice*.

More specifically, Behavior Ontology provides the meaning interpretation of the strong and weak equivalences since it is based on the *n:2-Lattice* mathematical structure. In addition, Behavior Ontology overcomes the complexity and non-deterministic conditions of all the interactions among the actors of the behaviors since the complexities and non-deterministic conditions are abstracted in the behavior patterns of the ontology.

In addition, in order to prove the concept of the method, two working tools were developed based on the ADOxx Meta-Modeling Platform: SAVE for dTP-Calculus and PRISM for Behavior Ontology.

The method and tools can be considered one of the most challenging research topics in the area of a domain engineering method to abstract the collective behavior of Smart IoT Systems.

The most interesting future research will focus on the probable similarity between two systems with respect to a set of identical processes or IoTs with some acceptable probability threshold.

## Figures and Tables

**Figure 1 sensors-22-05057-f001:**
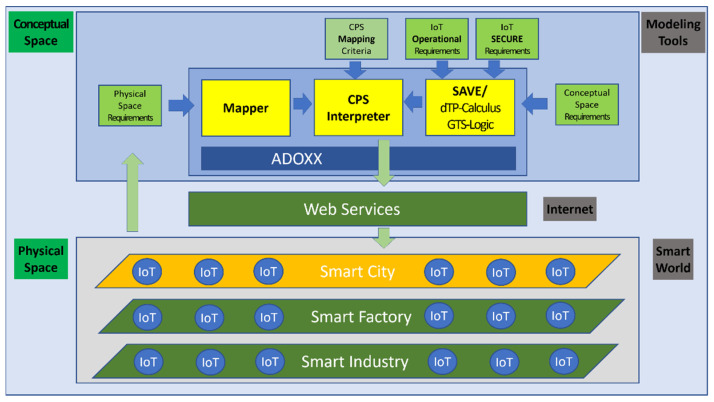
CPS Based on IoT Systems.

**Figure 2 sensors-22-05057-f002:**
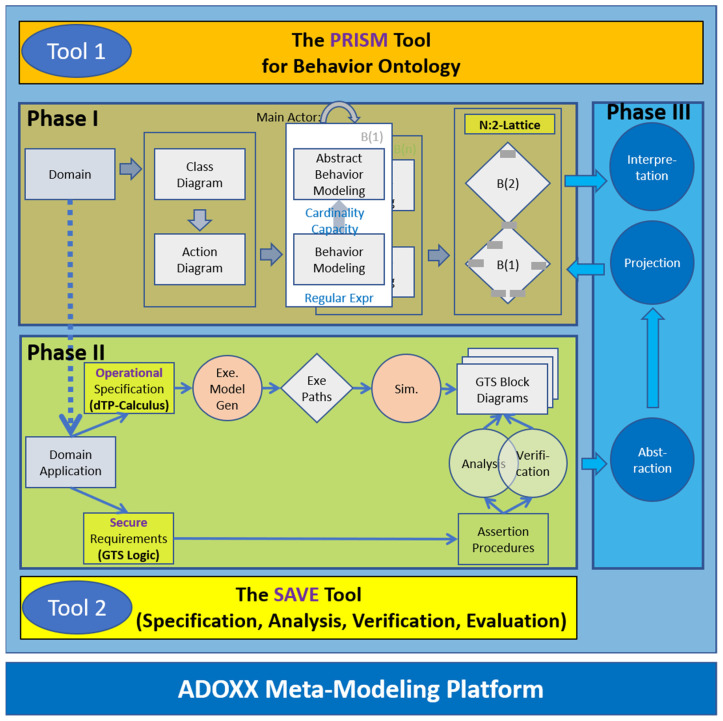
Approach.

**Figure 3 sensors-22-05057-f003:**
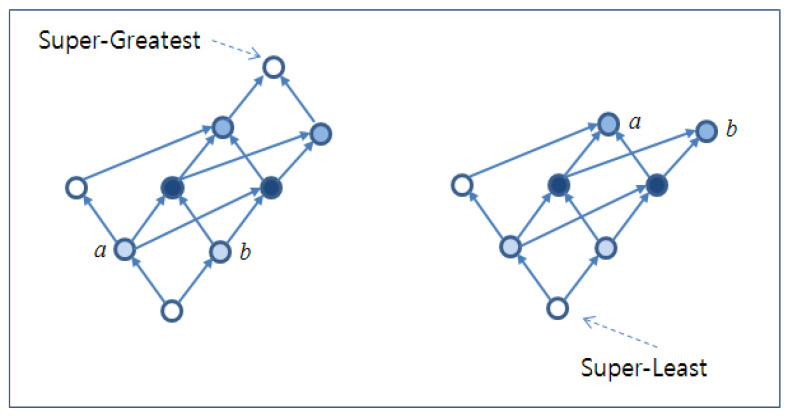
Examples for SGE and SLE.

**Figure 4 sensors-22-05057-f004:**
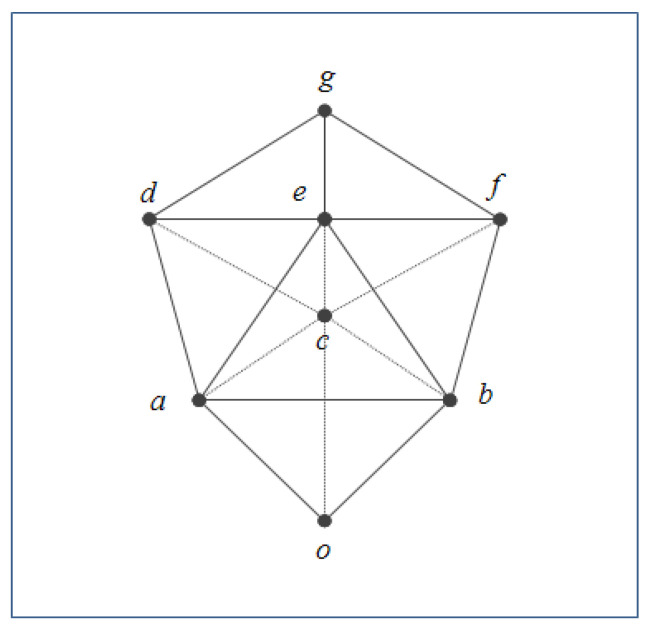
An Example for *n:2-Lattice*.

**Figure 5 sensors-22-05057-f005:**
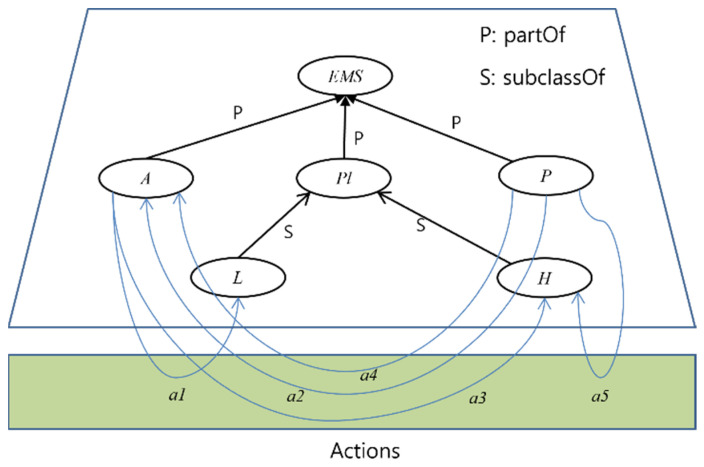
Active Ontology for Smart EMS Domain.

**Figure 6 sensors-22-05057-f006:**
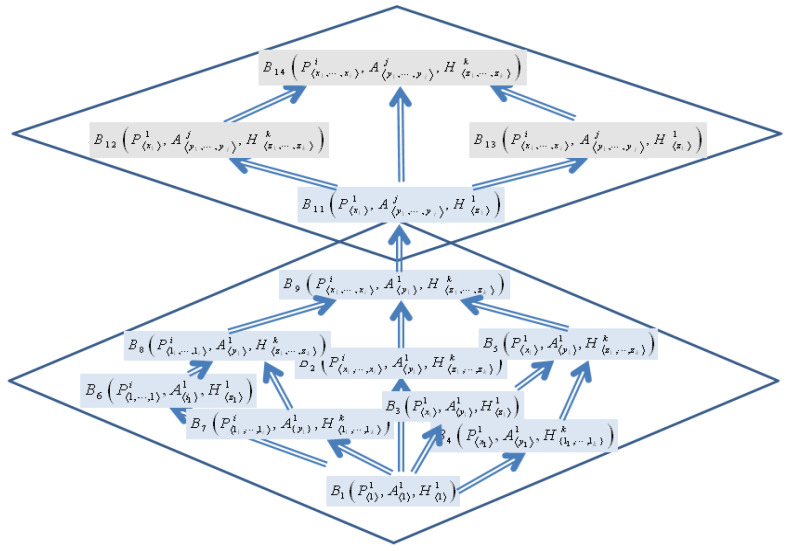
Behavior Lattices for EMS B(n, 1, n, n) and B(n, n, n, n).

**Figure 7 sensors-22-05057-f007:**
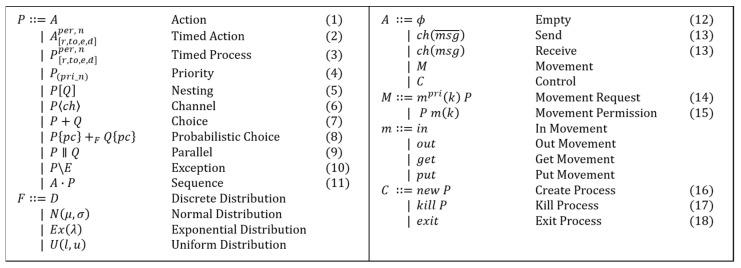
Syntax of dTP-Calculus.

**Figure 8 sensors-22-05057-f008:**
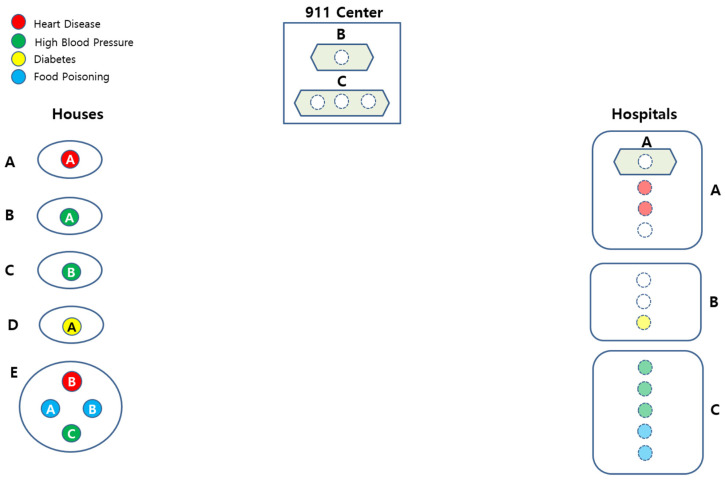
Configuration of the EMS Example.

**Figure 9 sensors-22-05057-f009:**
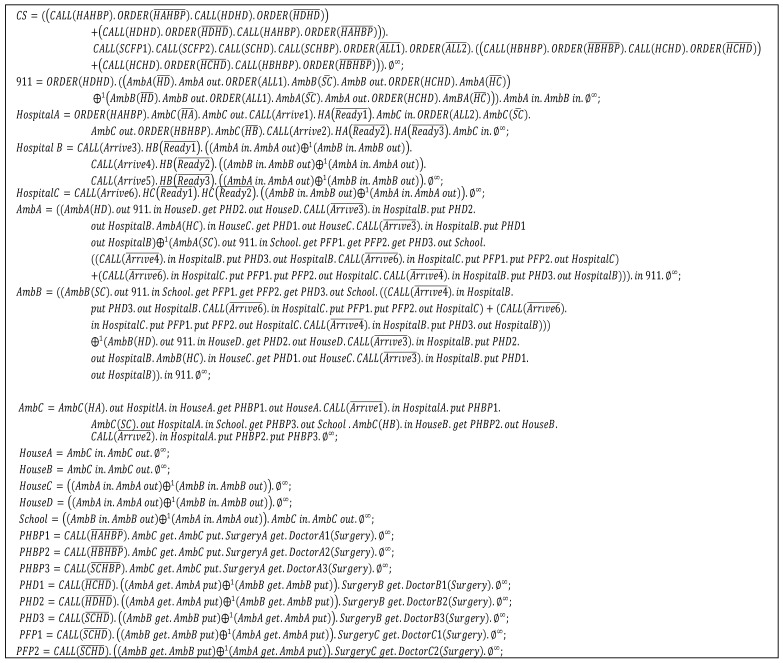
dTP-Calculus Code for the Smart EMS Example.

**Figure 10 sensors-22-05057-f010:**
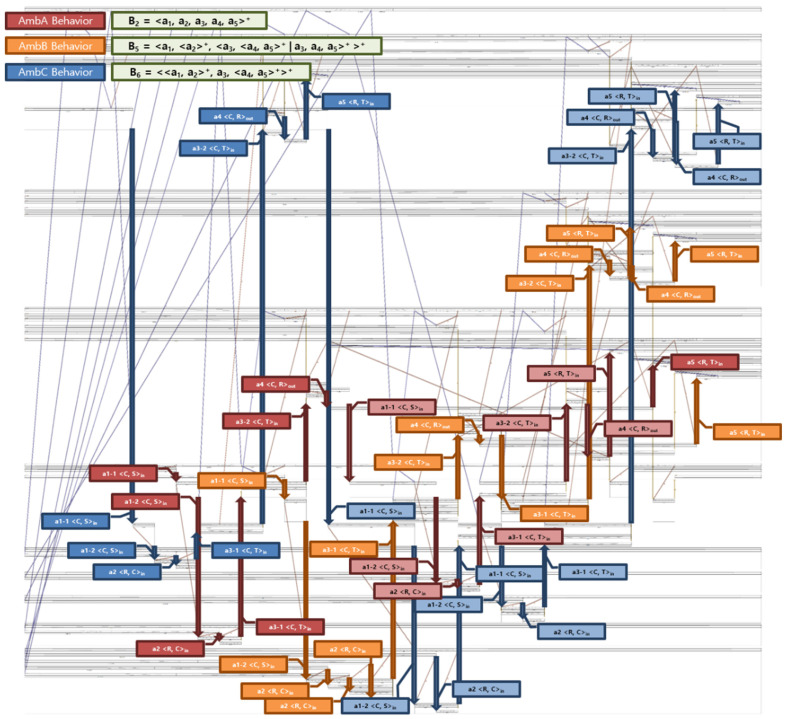
Simulation Output for EMS Example with Instances of Actions and Behaviors.

**Figure 11 sensors-22-05057-f011:**
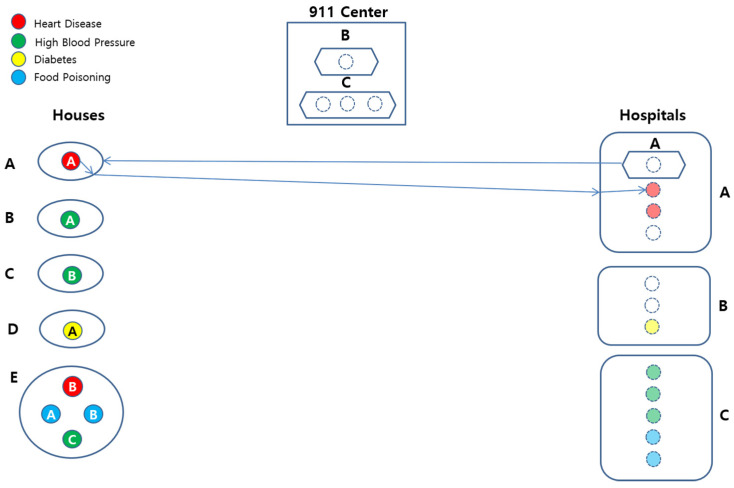
Graphical Representation of $B_{1.1}\left(P_{A^{H}}^{A},A_{\left\langle 1\right\rangle }^{A},H_{\left\langle 1\right\rangle }^{A}\right)$.

**Figure 12 sensors-22-05057-f012:**
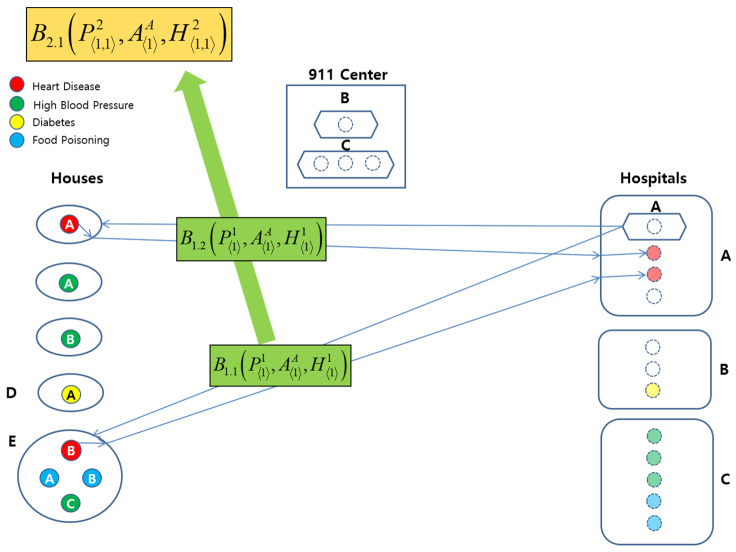
Graphical Representation of $B_{2.1}\left(P_{\left\langle 1,1\right\rangle }^{2},A_{\left\langle 1\right\rangle }^{A},H_{\left\langle 1,1\right\rangle }^{2}\right)$.

**Figure 13 sensors-22-05057-f013:**
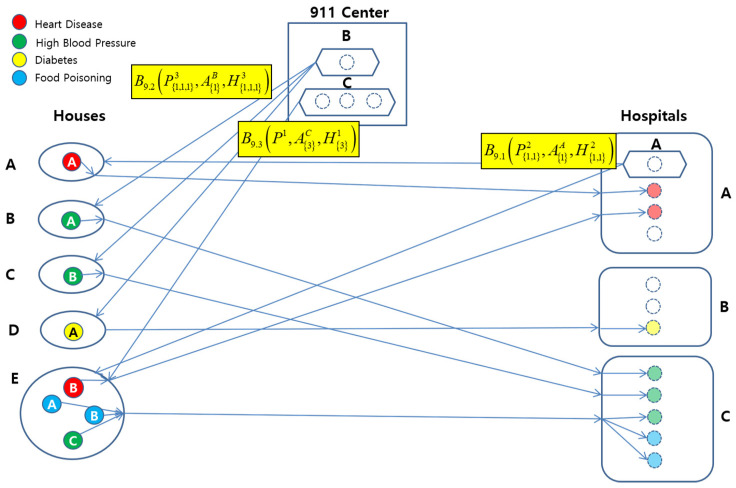
Graphical Representation of $B_{14}\left(P_{\left\langle 1,1,1,1,4\right\rangle }^{5},A_{\left\langle 1,1,3\right\rangle }^{3},H_{\left\langle 2,1,5\right\rangle }^{3}\right)$.

**Figure 14 sensors-22-05057-f014:**
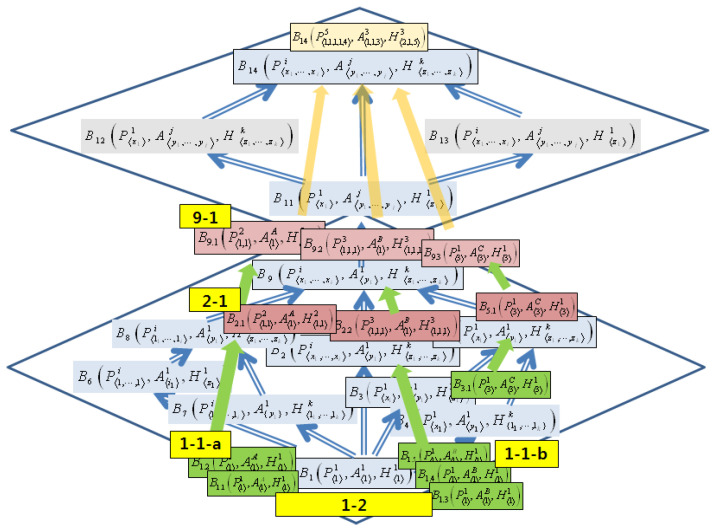
Projection of Behavior Instances to Behavior Ontology.

**Figure 15 sensors-22-05057-f015:**
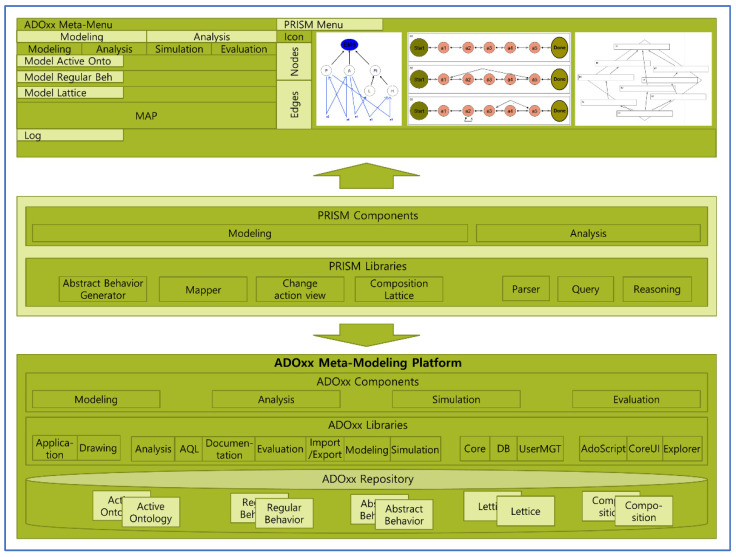
The architecture with modeling views of PRISM.

**Figure 16 sensors-22-05057-f016:**
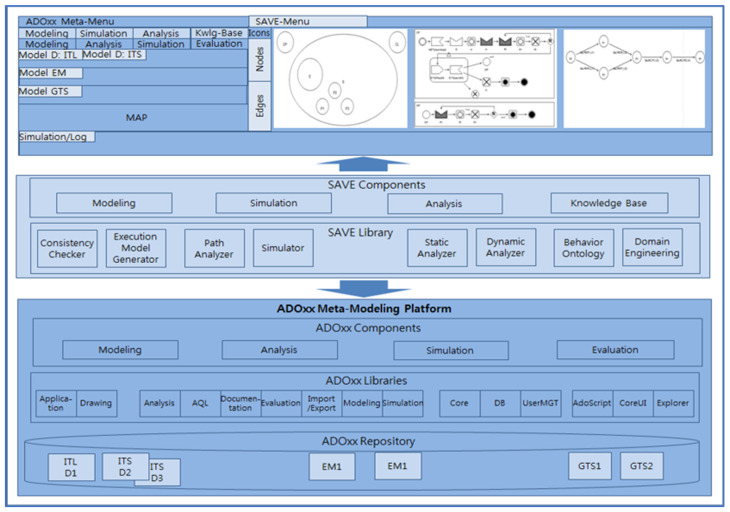
The architecture with modeling views of SAVE.

**Figure 17 sensors-22-05057-f017:**
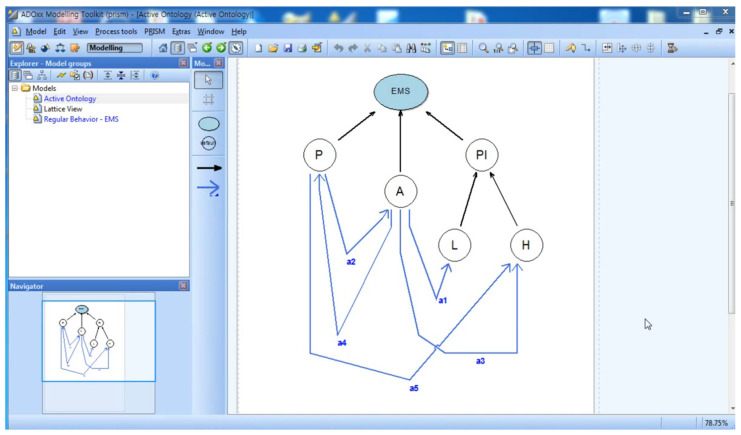
Phase I: Step 1.

**Figure 18 sensors-22-05057-f018:**
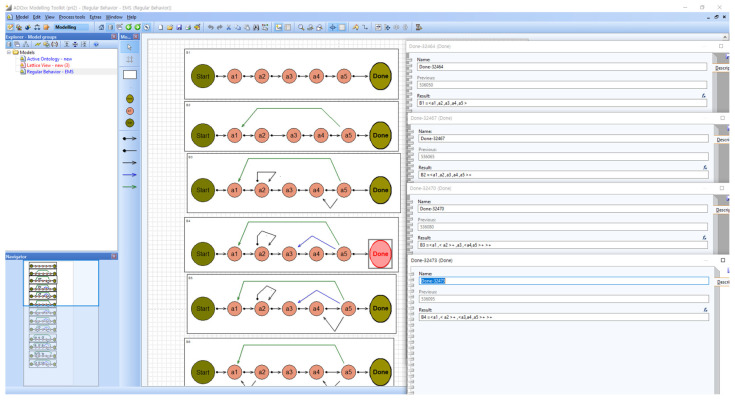
Phase I: Step 2.

**Figure 19 sensors-22-05057-f019:**
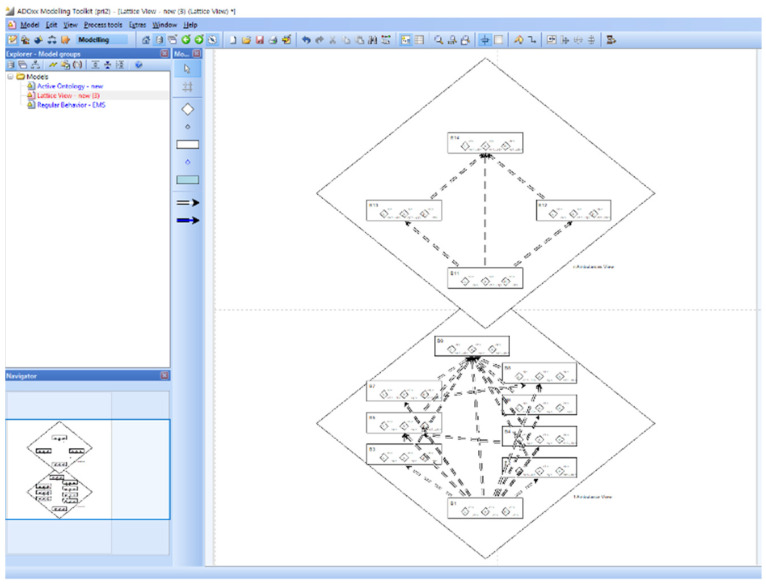
Phase I: Step 3.

**Figure 20 sensors-22-05057-f020:**
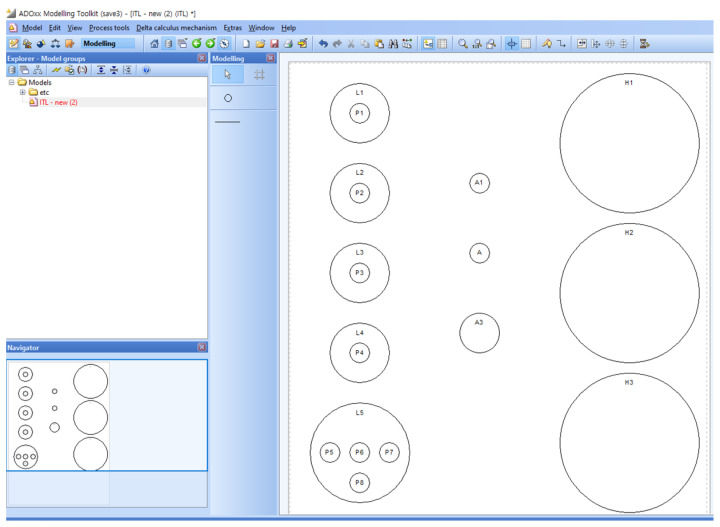
Phase II: Step 1.

**Figure 21 sensors-22-05057-f021:**
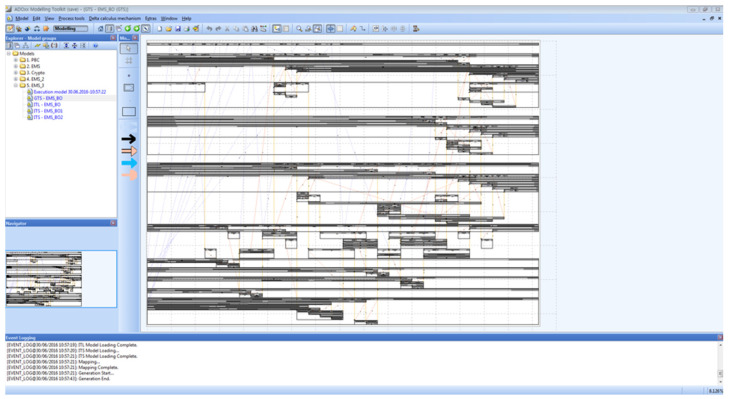
Phase II: Step 2.

**Figure 22 sensors-22-05057-f022:**
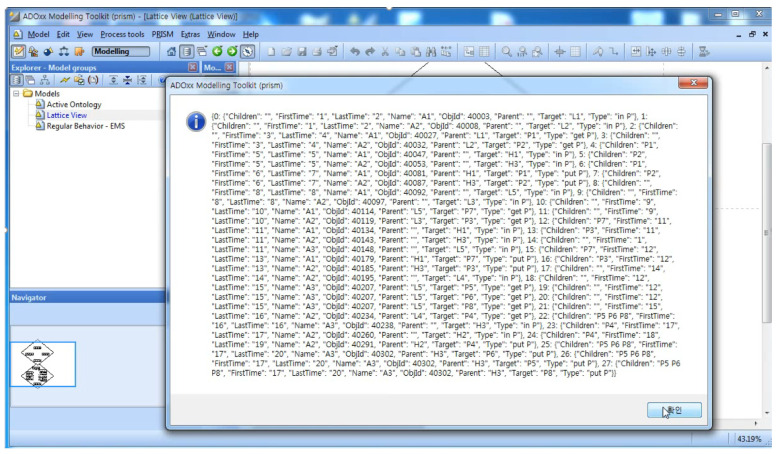
The Raw Data for Behaviors from Simulation in SAVE (Korean means confirm).

**Figure 23 sensors-22-05057-f023:**
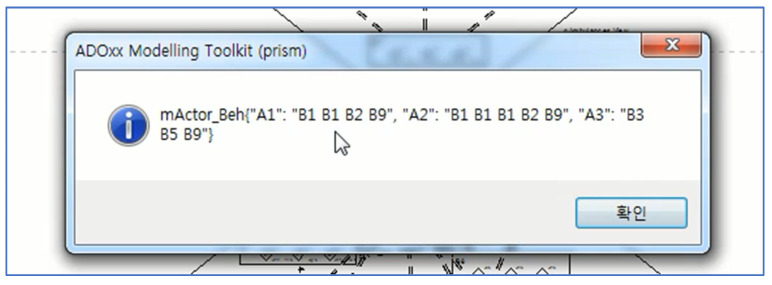
The Abstract Behaviors for Those in Figure 22 (Korean means confirm).

**Figure 24 sensors-22-05057-f024:**
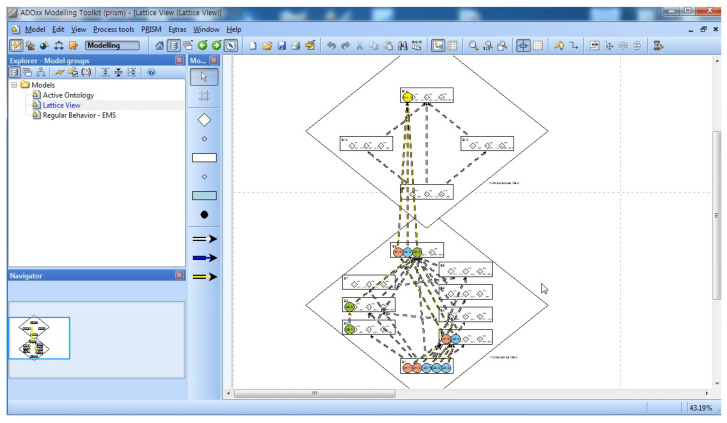
Phase III: Step 1.

**Figure 25 sensors-22-05057-f025:**
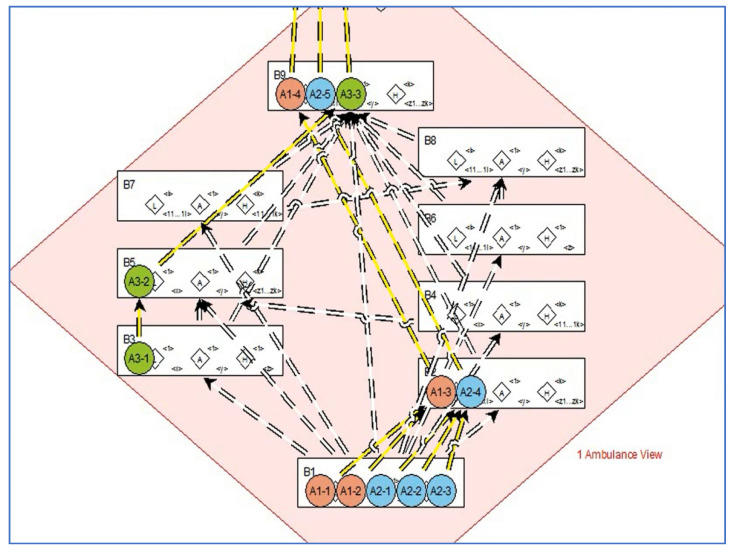
Phase III: Step 2.

**Table 1 sensors-22-05057-t001:** The comparative analysis of dTP-Calculus with other process algebra.

	PCCS	PACSR	pCCPS	PALOMA	dTP-Calculus
Distributivity	No	No	N/A	Agent	Geo-Space
Communication	τ-action	τ-action	τ-action	N/A	τ-action Synch Asyncho
Mobility	N/A	N/A	N/A	Agent	λ-action Active Passive
Real-Time	N/A	N/A	Time	N/A	Time
Probability	Conditional	Conditional	Discrete Distribution	Exponential Distribution	Discrete, Normal, Exponential, Uniform Distributions

**Table 2 sensors-22-05057-t002:** Type of Movements.

	Mode	Active	Passive
Direction			
Move-in		In	Get
Move-out		Out	Put

**Table 3 sensors-22-05057-t003:** Semantics of dTP-Calculus.

No	Name	Transition Rules
**(1)**	*Sequence*	$\frac{-}{A\cdot P\overset{\;A\;}{\to }P}$
**(2)**	*ChoiceL*	$\frac{-}{P+Q\overset{\;\;}{\to }P}$
	*ChoiceR*	$\frac{-}{P+Q\overset{\;\;}{\to }Q}$
**(3)**	*Probability Choice*	$\frac{A\cdot P\overset{A}{\to }\;P}{(\sum_{i\in I}A_{i}\left\{pc_{i}\right\})\cdot P\overset{A_{i}\left\{pc_{i}\right\}}{\to }P^{\prime }}\;\left(\underset{i\in I}{\sum }pc_{i}=1,\;i\in I\right)$
**(4)**	*Com*	$\frac{-}{ch_{1}\left(\bar{msg_{1}}\right)\cdot P\left|\left|ch_{2}\left(msg_{2}\right)\cdot Q\overset{\;\tau \;}{\to }P\right|\right|Q}\;\left(\left(ch_{1}=ch_{2}\right)\wedge \left(msg_{1}=msg_{2}\right)\right)$
**(5)**	*ParallelL*	$\frac{P\overset{\;\;}{\to }P^{\prime }}{P\left|\left|Q\overset{\;\;}{\to }P^{\prime }\right|\right|Q}$
	*ParallelR*	$\frac{Q\overset{\;\;}{\to }Q^{\prime }}{P\left|\left|Q\overset{\;\;}{\to }P\right|\right|Q^{\prime }}$
	*ParallelCom*	$\frac{P\overset{\;A\;}{\to }P^{\prime },Q\overset{\;\bar{A}\;}{\to }Q^{\prime }}{P\left|\left|Q\overset{\;\tau \;}{\to }P^{\prime }\right|\right|Q^{\prime }}$
**(6)**	*NestingO*	$\frac{P\overset{\;\;}{\to }P^{\prime }}{P\left[Q\right]\overset{\;\;}{\to }P^{\prime }\left[Q\right]}$
	*NestingI*	$\frac{Q\overset{\;\;}{\to }Q^{\prime }}{P\left[Q\right]\overset{\;\;}{\to }P\left[Q^{\prime }\right]}$
	*NestingCom*	$\frac{P\overset{\;A\;}{\to }P^{\prime },Q\overset{\;\bar{A}\;}{\to }Q^{\prime }}{P\left|\left|Q\overset{\;\tau \;}{\to }P^{\prime }\right|\right|Q^{\prime }}$
**(7)**	*In*	$\frac{P\overset{\;in\left(k\right)Q\;}{\to }P^{\prime },Q\overset{\;Pin\left(k\right)\;}{\to }Q^{\prime }}{P||Q\overset{\;\delta \;}{\to }Q^{\prime }\left[P^{\prime }\right]}$
	*Out*	$\frac{P\overset{\;out\left(k\right)Q\;}{\to }P^{\prime },Q\overset{\;Pout\left(k\right)\;}{\to }Q^{\prime }}{Q\left[P\right]\overset{\;\delta \;}{\to }P^{\prime }||Q^{\prime }}$
	*Get*	$\frac{P\overset{\;get\left(k\right)Q\;}{\to }P^{\prime },Q\overset{\;Pget\left(k\right)\;}{\to }Q^{\prime }}{P||Q\overset{\;\delta \;}{\to }P^{\prime }\left[Q^{\prime }\right]}$
	*Put*	$\frac{P\overset{\;put\left(k\right)Q\;}{\to }P^{\prime },Q\overset{\;Pput\left(k\right)\;}{\to }Q^{\prime }}{P\left[Q\right]\overset{\;\delta \;}{\to }P^{\prime }||Q^{\prime }}$
**(8)**	*InP*	$\frac{P\overset{\;in^{pri}\left(k\right)Q\;}{\to }P^{\prime }}{P_{\left(n_{1}\right)}||Q_{\left(n_{2}\right)}\overset{\;\delta \;}{\to }Q_{\left(n_{2}\right)}\left[{P^{\prime }}_{\left(n_{1}\right)}\right]}((n_{1}>n_{2}\wedge n_{2}\neq 0)\vee \left(n_{1}=0\wedge n_{2}\neq 0\right))$
	*OutP*	$\frac{P\overset{\;out^{pri}\left(k\right)Q\;}{\to }P^{\prime }}{Q_{\left(n_{2}\right)}\left[P_{\left(n_{1}\right)}\right]\overset{\;\delta \;}{\to }{P^{\prime }}_{\left(n_{1}\right)}||Q_{\left(n_{2}\right)}}((n_{1}>n_{2}\wedge n_{2}\neq 0)\vee \left(n_{1}=0\wedge n_{2}\neq 0\right))$
	*GetP*	$\frac{P\overset{\;get^{pri}\left(k\right)Q\;}{\to }P^{\prime }}{P_{\left(n_{1}\right)}||Q_{\left(n_{2}\right)}\overset{\;\delta \;}{\to }{P^{\prime }}_{\left(n_{1}\right)}\left[Q_{\left(n_{2}\right)}\right]}((n_{1}>n_{2}\wedge n_{2}\neq 0)\vee \left(n_{1}=0\wedge n_{2}\neq 0\right))$
	*PutP*	$\frac{P\overset{\;put^{pri}\left(k\right)Q\;}{\to }P^{\prime }}{P_{\left(n_{1}\right)}\left[Q_{\left(n_{2}\right)}\right]\overset{\;\delta \;}{\to }{P^{\prime }}_{\left(n_{1}\right)}||Q_{\left(n_{2}\right)}}((n_{1}>n_{2}\wedge n_{2}\neq 0)\vee \left(n_{1}=0\wedge n_{2}\neq 0\right))$
**(9)**	*TickTimeR*	$\frac{-}{A_{\left[r,to,e,d\right]}^{per,n}\cdot P\overset{\;⊳1\;}{\to }A_{\left[r-1,to,e,d-1\right]}^{per,n}\cdot P}\left(r\geq 1\right)$
**(10)**	*TickTimeTO*	$\frac{A\cdot P\left|\left|\bar{A}\cdot Q\overset{\;\tau \vee \delta \;}{\to }P\right|\right|Q}{A_{\left[0,to,e,d\right]}^{per,n}\cdot P\overset{\;⊳1\;}{\to }A_{\left[0,to-1,e,d-1\right]}^{per,n}\cdot P}\left(to\geq 1\right)$
**(11)**	*TickTimeSyncE*	$\frac{A\cdot P\left|\left|\bar{A}\cdot Q\overset{\;\tau \vee \delta \;}{\to }P\right|\right|Q}{A_{\left[0,to_{1},e_{1},d_{1}\right]}^{per_{1},\;\;\;n_{1}}\cdot P\left|\left|{\bar{A}}_{\left[0,to_{2},e_{2},d_{2}\right]}^{per_{2},\;\;\;n_{2}}\cdot Q\overset{\;⊳1\;}{\to }A_{\left[0,to_{1},e_{1}-1,d_{1}-1\right]}^{per_{1},\;\;\;n_{1}}\cdot P\right|\right|{\bar{A}}_{\left[0,to_{2},e_{2}-1,d_{2}-1\right]}^{per_{2},\;\;\;n_{2}}\cdot Q}\left(e_{1}\geq 1,e_{2}\geq 1\right)$
**(12)**	*TickTimeAsyncE*	$\frac{-}{A_{\left[0,to,e,d\right]}^{per,n}\cdot P\overset{\;⊳1\;}{\to }A_{\left[0,to,e-1,d-1\right]}^{per,n}\cdot P}\left(e\geq 1\right)$
**(13)**	*TickTimeEnd*	$\frac{-}{A_{\left[0,to,0,d\right]}^{per,n}\cdot P\overset{\;⊳1\;}{\to }P}$
**(14)**	*Timeout*	$\frac{-}{\left(A_{\left[0,\;0,e,d\right]}^{per,n}\backslash E\right)\cdot P\overset{\;⊳1\;}{\to }E\cdot P}$
**(15)**	*Deadline*	$\frac{-}{\left(A_{\left[r,\;to,e,0\right]}^{per,n}\backslash E\right)\cdot P\overset{\;⊳1\;}{\to }E\cdot P}$
**(16)**	*Period*	$\frac{-}{A_{\left[r,to,e,d\right]}^{per,n}\cdot P\overset{\;⊳per\;}{\to }A_{\left[r,to,e,d\right]}^{per,\;\;\;n-1}\cdot P}\left(n\geq 1\right)$
**(17)**	*Period End*	$\frac{-}{A_{\left[0,to,0,d\right]}^{per,0}\cdot P\overset{\;⊳1\;}{\to }P}$

## Data Availability

Not applicable.

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
