# Peer review of "Modeling Method to Abstract Collective Behavior of Smart IoT Systems in CPS"

_sensors, 2022, doi:10.3390/s22135057_

Round 1

Reviewer 1 Report

The Abstract of the manuscript-at-hand should be delineated in a much more categorical manner, i.e., it should succinctly highlight the overall domain, the challenges of the said domain, and the salient challenge addressed here in this particular manuscript. On the contrary, the authors have delineated the entire approach, i.e., in the form of a number of steps and which is, in fact, something to be delineated in the section, Methodology, and not the Abstract.

The Introduction should be presented in a logical order, i.e., from the evolution of the IoT to the Smart IoT followed by the CPS. The authors may like to study the following IoT literature, i.e., https://doi.org/10.1109/CIC50333.2020.00015 and https://doi.org/10.1145/3464960, or any similar literature in this regard.

How about the state-of-the-art? A separate section entitled, Related Work or State-of-the-Art, should be introduced delineating both the pros and cons of the existing relevant literature from within the past three, or at the most, past five years (with the exception of the Seminal Works). The ones highlighted in the Bibliography are pretty outdated with a couple of them being from 1993 and 1996. 

The quality of Figures 10, 21, 22, and 25 (in particular) and of Figures 15, 16, 17, 20, and 23 need to enhanced. Unnecessary Figures may be removed in a bid to control the length of the manuscript, i.e., too long manuscript in most of the cases also lose its audience.

Reviewer 2 Report

The authors propose a knowledge-based architecture for smart IoT systems. The paper is well-written, easy to follow even with very dense mathematical concepts, and clear in its goals. Overall, a few points need to be addressed before publication:

1) The introduction is quite weak in clarifying why this proposed architecture is necessary from an academic perspective. Why does current CPS architectures are not sufficient? What does this knowledge-based architecture provide that has not been done before? The need for this paper is not clear from the beginning due to the lack of a proper lit. review. Also, it is hard to conceptualize the contributions of the authors considering that a lot of the written content seems to be taken directly from the authors' previous works.

2) Figure captions should be way more descriptive (e.g., figure 2).

3) The rest of points to clarify will depend on the authors changes based on point 1. For example, this reviewer wonders why the behaviour ontology is required to be modelled with a lattice structure. Is that always the case or is it just because of the ontology structure shown in Fig.5 that coincides with a n-2 lattice? Can it be generalized?

4) One of the benefits of ontologies is interoperability. How does the extension of any subclass impact the lattice structure reasoning? Will it need to be reevaluated? If so, why is this a better solution that using current ontology models in OWL format for example?

Round 2

Reviewer 1 Report

Thank you for addressing the comments.

The quality of the manuscript-at-hand has considerably improved.